

# Seismic reflection data reveal the 3D structure of the newly discovered Exmouth Dyke Swarm, offshore NW Australia

Craig Magee[1], Christopher A-L Jackson[2]

[1]Institute of Geophysics and Tectonics, School of Earth Science and Environment, University of Leeds, Leeds, LS2 9JT, UK
[2]Basins Research Group (BRG), Department of Earth Science and Engineering, Imperial College London, London, SW7 2BP, UK

*Correspondence to*: Craig Magee (c.magee@leeds.ac.uk)

**Abstract.** Dyke swarms are common on Earth and other planetary bodies, comprising arrays of dykes that can extend for 10's to 1000's of kilometres. The vast extent of such dyke swarms, and their rapid emplacement, means they can significantly influence a variety of planetary processes, including continental break-up, crustal extension, resource accumulation, and volcanism. Determining the mechanisms driving dyke swarm emplacement is thus critical to a range of Earth Science disciplines. However, unravelling dyke swarm emplacement mechanics relies on constraining their 3D structure, which is extremely difficult given we typically cannot access their subsurface geometry at a sufficiently high enough resolution. Here we use high-quality seismic reflection data to identify and examine the 3D geometry of the newly discovered Exmouth Dyke Swarm, and associated structures (i.e. dyke-induced normal faults and pit craters), in unprecedented detail. The latest Jurassic dyke swarm is located on the Gascoyne Margin offshore NW Australia and contains numerous dykes that are >170 km long, potentially >500 km long. The mapped dykes are distributed radially across a 39° arc centred on the Cuvier Margin; we infer this focal area marks the source of the dyke swarm, which was likely a mantle plume. We demonstrate seismic reflection data provides unique opportunities to map and quantify dyke swarms in 3D in sedimentary basins, which can allow us to: (i) recognise dyke swarms across continental margins worldwide and incorporate them into models of basin evolution and fluid flow; (ii) test previous models and hypotheses concerning the 3D structure of dyke swarms; (iii) reveal how dyke-induced normal faults and pit craters relate to dyking; and (iv) unravel how dyking translates into surface deformation.

## 1 Introduction

Dyke swarm emplacement can rapidly transfer large volumes of magma through the crust, over 10's to 1000's of kilometres, on Earth and on other planetary bodies (e.g., Fig. 1a) (Ernst, 2014; Bryan and Ernst, 2008; Coffin and Eldholm, 2005; Coffin and Eldholm, 1994; Halls and Fahrig, 1987; Halls, 1982; Ernst and Baragar, 1992; Wilson and Head, 2002). We recognise three principal dyke swarm geometries: (i) *parallel or linear dyke swarms*, which typically develop orthogonal to a far-field $\sigma_3$, within and sub-parallel to rift zones (e.g., Fig. 1b) (e.g., Ernst et al., 2001; Paquet et al., 2007; Ebinger and Casey, 2001); (ii) *radial dyke swarms*, which form when $\sigma_3$ is circumferential to a large volcanic centre or mantle plume source (e.g., Figs





1a and c) (e.g., Buchan and Ernst, 2013; Baragar et al., 1996; Odé, 1957; Walker, 1986); and (iii) *circumferential dyke swarms*, which likely emanate from the lateral termination of a plume head, although the stress state controlling their emplacement remains poorly understood (e.g., Fig. 1d) (e.g., Buchan and Ernst, 2018a, b). Component dykes within dyke swarms can be up to 10's or 100's m thick and their emplacement is primarily accommodated by extending the host rock

(e.g., Jolly and Sanderson, 1995; Rivalta et al., 2015; Gudmundsson, 1983; Paquet et al., 2007). Their geometry and scale means dyke swarms can thus drive crustal extension, influencing plate tectonic processes on Earth and shaping other planetary bodies (e.g., Ernst et al., 2013; Ernst and Buchan, 1997; Wilson and Head, 2002; Ebinger and Casey, 2001; Halls, 1982; Paquet et al., 2007). Because they are typically emplaced over short timespans (≲5 Myr) and are sensitive to the prevailing stress field, dyke swarms also provide a record of syn-emplacement stress conditions and represent key spatial and

temporal markers for palaeogeographic and palinspastic reconstruction (e.g., Hou et al., 2010; Bleeker and Ernst, 2006; Ju et al., 2013; Peng, 2015; Halls, 1982). Furthermore, dyke swarms may be associated with the accumulation of critical economic resources (e.g., Jowitt et al., 2014; Ernst and Jowitt, 2013) and, if they feed extensive flood basalts, may contribute to climate change and related mass extinctions (e.g., Ernst and Youbi, 2017). Unravelling the emplacement history of dyke swarms and deciphering the processes controlling their intrusion and form, is therefore crucial to a wide range of pure and applied Earth

Science disciplines.

Decoding dyke swarm emplacement requires knowledge of their 3D structure, which is typically inferred by quantifying and projecting downwards the plan-view morphology of dykes exposed at Earth's surface or identified in airborne/satellite imagery (e.g., Ernst and Youbi, 2017; Ernst, 2014; Bryan and Ernst, 2008; Coffin and Eldholm, 2005; Coffin and Eldholm, 1994; Bryan et al., 2010; Hou et al., 2010; Halls and Fahrig, 1987; Halls, 1982; Ernst and Baragar,

1992). Such inferences of 3D structure may be augmented by direct mapping of the local subsurface structure of dyke swarms, or component dykes, intersected in mines or imaged in geophysical data (e.g., Kavanagh and Sparks, 2011; Keir et al., 2011; Wall et al., 2010). Whilst integrating these datasets has emphasised the lateral variability in dyke swarm architecture, a recent seismic reflection-based study has highlighted dyke swarm structure can also change with depth (Phillips et al., 2018). In particular, Phillips et al. (2018) demonstrated the width of a dyke swarm imaged offshore southern

Norway increased with depth, implying the plan-view morphology of a dyke swarm may not be a proxy for its 3D geometry (or total volume); i.e. the plan-view morphology of a dyke swarm is a function of its attitude relative to the present topography (see also Magee et al., 2019). We can use different physical, analytical, and numerical modelling approaches to evaluate the 3D geometry of dyke swarms, and to establish how their structure can be inferred from principally 2D, surface-based analyses. However, these model predictions are difficult to validate without constraints on the true 3D form of natural

dyke swarms (e.g., Paquet et al., 2007; Bunger et al., 2013; Jolly and Sanderson, 1995; Macdonald et al., 1988). Advancing our understanding of dyke swarm emplacement thus requires a method for imaging their 3D structure in detail (e.g., Magee et al., 2019; Magee et al., 2018; Phillips et al., 2018).

Reflection seismology has proved a powerful tool for imaging the 3D structure of magma plumbing systems (see Magee et al., 2018 and references therein). Yet vertical dykes are commonly expressed as very subtle reflection





discontinuities within seismic reflection data, and are thus easily and often overlooked (e.g., Fig. 2) (e.g., Wall et al., 2010;
      Bosworth et al., 2015; Malehmir et al., 2018; Holford et al., 2017; Kirton and Donato, 1985; Ardakani et al., 2017; Jaunich,
      1983; Plazibat et al., 2019). Whilst dykes have been recognised in seismic reflection data (e.g., Fig. 2), we are not aware of
      any concerted effort to quantify their 3D geometry across large areas (>10's of kilometres) using this technique. Here, we
      use an extensive suite of 2D and 3D seismic reflection data from the North Carnarvon Basin, offshore NW Australia to

examine the 3D structure of a previously unidentified dyke swarm, which we name the Exmouth Dyke Swarm. We aim to:
      (i) characterise the dyke swarms seismic expression and identify diagnostic criteria that can be used to identify dykes in
      other seismic reflection datasets; (ii) quantify dyke geometry (e.g., length and spacing) and test predictions of how dyke
      populations develop in time and space; and (iii) decipher the tectono-magmatic and geodynamic setting of the Exmouth
      Dyke Swarm.

**2 Geological Setting**

      The North Carnarvon Basin is located on the ~500 km wide, magma-rich Gascoyne Margin, offshore NW Australia (Fig.
      3a). The basin extends southward onto the ~100–150 km wide Cuvier Margin, which is separated from the Gascoyne Margin
      by the Cape Range Fracture Zone (Fig. 3a). Tectonic elements within the North Carnarvon Basin include the Exmouth
      Plateau, the Exmouth, Barrow, and Dampier sub-basins, and the Carnarvon Terrace (Fig. 3a). Basin formation involved

several episodic rifting events between the Late Carboniferous and Early Cretaceous, with sub-basin development initiating
      in the Late Triassic (Fig. 3b) (e.g., Stagg and Colwell, 1994; Longley et al., 2002; Tindale et al., 1998; Willcox and Exon,
      1976; Jitmahantakul and McClay, 2013; Gartrell et al., 2016; Black et al., 2017). This Late Triassic rifting continued until
      the near end Callovian (~164 Ma), when extension was interrupted by a phase of regional uplift recorded in the formation of
      a major unconformity (Fig. 3b) (e.g., Tindale et al., 1998; Jitmahantakul and McClay, 2013; Gartrell et al., 2016). Renewed

extension in the Late Jurassic-to-Early Cretaceous, which likely initiated in the Tithonian, occurred in response to rifting
      between Greater India and Australia (Fig. 3b) (e.g., Magee et al., 2016a; Longley et al., 2002; Stagg et al., 2004; Tindale et
      al., 1998). Rifting during the Early Cretaceous involved discrete periods of unconformity development and culminated in
      continental break-up at ~130 Ma during the Hauterivian (Figs 3a and b) (e.g., Robb et al., 2005; Direen et al., 2008; Heine
      and Müller, 2005; Willcox and Exon, 1976; Stagg et al., 2004). Following continental break-up, post-rift thermal subsidence

has controlled passive margin evolution (e.g., Kaiko and Tait, 2001; Jitmahantakul and McClay, 2013; Tindale et al., 1998).
      During the post-rift period, several tiers of polygonal fault systems developed across much of the North Carnarvon Basin
      (e.g., Velayatham et al., 2019).

      **2.1 Stratigraphic framework**

      Sedimentary sequences within the North Carnarvon Basin are typically 10–18 km thick, and locally up to 24 km thick in the

sub-basins, making it difficult to seismically image the <10 km thick crystalline basement (e.g., Fig. 3c) (e.g., Reeve et al.,





2016; Tindale et al., 1998; Stagg et al., 2004; Stagg and Colwell, 1994; Mutter and Larson, 1989). Borehole data show the dyke-hosting interval of interest comprises (Figs 3b and c): (i) siliciclastic rocks of the Late Permian-to-Late Triassic marine Locker Shale and fluvio-deltaic Mungaroo Formation, which are up to 9 km thick (e.g., Stagg et al., 2004; Longley et al., 2002; Hocking et al., 1987; Tindale et al., 1998); (ii) Late Triassic-to-Late Jurassic marine claystones and marls (i.e. the

Brigadier and North Rankin formations, Murat Siltstone, Athol Formation, and Dingo Claystone), which are up to 4 km thick in the Barrow and Exmouth sub-basins but only preserved as a condensed, <100 m thick succession on the Exmouth Plateau (e.g., Tindale et al., 1998; Stagg et al., 2004; Stagg and Colwell, 1994; Hocking, 1992; Jitmahantakul and McClay, 2013); and (iii) Late Jurassic-to-Early Cretaceous (Tithonian-to-Valanginian; ~146.7–138.2 Ma) clastic deltaic rocks of the Barrow Group and the overlying coastal Birdrong Sandstone (e.g., Reeve et al., 2016; Paumard et al., 2018). Late Jurassic-to-Early

Cretaceous, rift-related unconformities have, in places, eroded down into the Mungaroo Formation (Fig. 3c) (e.g., Reeve et al., 2016).

## 2.2 Mesozoic tectonic faulting

Mesozoic extension produced two principal fault arrays in the North Carnarvon Basin. Late Triassic-to-Middle Jurassic rifting led to development of NE-SW striking, domino-style normal faults that have >1 km of throw (e.g., Fig. 3c) (e.g.,

Jitmahantakul and McClay, 2013; Magee et al., 2016a; Black et al., 2017; Tindale et al., 1998). Late Jurassic-to-Early Cretaceous rifting was characterised by the formation of broadly NE-SW striking, low-throw (<0.1 km) normal faults that are primarily strata-bound between the Callovian and near Base Cretaceous or Valanginian unconformities (e.g., Tindale et al., 1998; Magee et al., 2016a; Jitmahantakul and McClay, 2013; Black et al., 2017). During the main period of Late Jurassic-to-Early Cretaceous rifting, as well as during younger faulting events (e.g., polygonal faulting), Late Triassic-to-

Middle Jurassic normal faults were locally reactivated (e.g., Jitmahantakul and McClay, 2013; Magee et al., 2016a). Stretching factors of $\beta$<1.2 for both Mesozoic rift events indicate the Exmouth Plateau accommodated only minor upper crustal extension during these periods (e.g., Driscoll and Karner, 1998; Bilal et al., 2018).

## 2.3 Magmatism

Igneous activity throughout the Late Jurassic-to-Early Cretaceous resulted in (Fig. 3b): (i) sill-complex emplacement, which

likely began in the Kimmeridgian prior to onset of rifting, across the Exmouth Plateau, Exmouth Sub-basin, and Carnarvon Terrace (e.g., Fig. 3a) (e.g., Symonds et al., 1998; Magee et al., 2013b; Holford et al., 2013; Magee et al., 2017; Magee et al., 2013a); (ii) intrusion of dykes, perhaps genetically related to sill intrusion (Rohrman, 2015); and (iii) development of a magma-rich, continent-ocean transition zone (COTZ) spanning the north-western edges of the Gascoyne and Cuvier margins in the Valanginian-to-Hauterivian (~136–130 Ma; Fig. 3a) (e.g., Rey et al., 2008; Symonds et al., 1998; Mihut and Müller,

1998; Direen et al., 2007). High-amplitude seismic reflections observed towards the base of the crust (Fig. 3c), coupled with a coincident downward increase in seismic velocity (from 6.2 km s$^{-1}$ to ~7.4 km s$^{-1}$), suggest igneous material was also emplaced in or below the lower crust during the Late-Jurassic-to-Early Cretaceous (~165–136 Ma) (Stagg et al., 2004;





Rohrman, 2013; Mutter and Larson, 1989; Frey et al., 1998). Previous studies have attributed this Late-Jurassic-to-Early Cretaceous magmatism to rift-related decompression melting (e.g., Karner and Driscoll, 1999), perhaps enhanced by small-

scale mantle convection (e.g., Hopper et al., 1992; Mutter et al., 1988; Mihut and Müller, 1998), and/or mantle plume activity (e.g., Rohrman, 2015, 2013; Müller et al., 2002; Black et al., 2017).

## 3 Dataset and methods

Dykes are rarely imaged in seismic reflection data because their sub-vertical orientation preferentially reflects seismic energy deeper into the subsurface, rather than returning it to the surface to be recorded (e.g., Eide et al., 2018; Thomson, 2007).

Dykes identified in the field and/or in aeromagnetic data have been indirectly recognised in co-located seismic reflection data where a localised reduction in returned seismic energy disrupts the continuity and strength (amplitude) of reflections associated with stratigraphic layering (e.g., Fig. 2) (e.g., Wall et al., 2010; Bosworth et al., 2015; Kirton and Donato, 1985; Ardakani et al., 2017); i.e. in these cases, dykes do not correspond to discrete reflections, but instead appear as 'vertical zones of disruption' (VZDs). Whilst dykes can thus be recognised in seismic reflection data, vertical strike-slip and normal

faults, and non-magmatic fluid flow conduits (e.g., gas chimneys) may also be expressed as VZDs. To avoid interpretational bias, we describe the features of interest in this study as VZDs, and collect additional data and make further observations to inform a critical discussion of their likely origin.

We use eight 3D and 63 2D, time-migrated seismic surveys to map 26 VZDs across ~40,000 km$^2$ of the North Carnarvon Basin (Figs 4a and b); the properties of each seismic survey are provided in Supplementary Table S1. Visual

inspection of the data and extraction of variance volume attributes, which highlight trace-to-trace variations in seismic wavelets to reveal structural (e.g., faults and VZDs) and stratigraphic (e.g. channel edges) discontinuities (Brown, 2011), allow us to identify VZDs in the 3D seismic volumes. These VZDs were mapped on sections oriented orthogonal to their strike every ~250–1200 m. In places, the VZDs were obscured by tectonic faults and could not be mapped at regular intervals. Along-strike projection of mapped VZDs outside of the 3D seismic volumes guided their interpretation on 2D

seismic lines, where poorer data quality and/or lower resolution hindered their recognition; we were able to confidently recognise VZDs in nine 2D seismic surveys (e.g., Figs 4c and d), although we cannot rule out their presence in other datasets.

In addition to mapping VZDs, we used biostratigraphic and well-log data from 24 wells to identify and interpret two key stratigraphic horizons across the study area: (i) the ~148 Myr near Base Cretaceous unconformity (BC); and (ii) the near

Top Mungaroo Formation (TM), which is broadly equivalent to the Norian-Rhaetian boundary (i.e. intra-Upper Triassic) (Fig. 4a; Supplementary Fig. S1). Where we observed fluvial channels within the Triassic strata using variance time-slices (e.g., Fig. 5), we locally mapped intra-Mungaroo horizons to assess channel continuity across identified VZDs; this helped us assess VZD kinematics. We also interpreted key structures associated with the VZDs, including overlying normal fault systems, pipes, and sub-circular depressions.





## 3.1 Quantitative analysis

The mechanics and dynamics of dyke swarm emplacement controls and is reflected in the geometry of its component dykes (e.g., Bunger et al., 2013; Jolly and Sanderson, 1995; Mège and Korme, 2004; Gudmundsson, 1987). For example, dyke lengths within a swarm are expected to display a power-law distribution and may be used to differentiate feeder and non-feeder dykes within a given population (Mège and Korme, 2004). The range of dyke thicknesses, which follows a Weibull distribution, and spacings can also provide insights into the strength of host rock and/or magma source conditions (Krumbholz et al., 2014; Bunger et al., 2013). These predicted distributions for dyke properties within a swarm also allow us to test whether an observed dyke set comprises one or multiple generations of intrusion, perhaps originating from different sources (e.g., Krumbholz et al., 2014). We quantify VZD structure and compare our results to predicted distributions to help unravel the mechanics and dynamics of VZD formation.

We measured the plan-view, tip-to-tip length ($L$) and strike ($S$) of each VZD (Fig. 5). Many VZDs display minor but abrupt changes in strike along their length (e.g., Fig. 5). These minor changes in strike sub-divide the VZDs into discrete planar segments, for which we measured strike ($s$) and length ($l$) (Fig. 5). Where coverage of 3D seismic volumes was sufficient, we also measured VZD width ($w$) and spacing ($h$) orthogonal to strike, on variance time-slices at 4.5 s two-way time (TWT) (Fig. 5); we specifically measured w and h, as well as the depth to VZD tips, along 35 ~E-W trending transects spaced ~4.7 km apart. Because data quality generally decreases with depth within individual seismic surveys, defining the base of individual VZDs is problematic, making it difficult to ascertain whether most VZDs truly terminate downwards or if they extend below the survey limits. We therefore only qualitatively assess VZD height ($H$).

## 3.2 Seismic resolution

We used time-depth plots derived from the checkshot data available for the 24 wells to estimate seismic velocities (Supplementary Fig. S3 and Supplementary Table S2). Because the VZDs extend below the total depth of all wells, we estimated seismic velocities ($v$) through the interval of interest by extrapolating a second-order polynomial trend-line through the cumulative checkshot data (Supplementary Fig. S3). The dominant frequency ($f$) of the 2D and 3D seismic surveys broadly decrease with depth from a maximum of ~30–40 Hz at the top of the interval of interest (~2.8–2.9 s TWT; ~2.5–2.7 km) to a minimum of ~5–20 Hz at ~5.9–6.0 s TWT (9.7–10.1 km). We calculated the average interval velocities for ~2.8–2.9 s TWT (~3.0 km s$^{-1}$) and ~5.9–6.0 s TWT (~6.4 km s$^{-1}$). Coupled with the dominant frequency data, these average interval velocities allowed us to estimate the dominant wavelength ($\lambda = v/f$) of the data and constrain the limits of separability (~$\lambda/4$) and visibility (~$\lambda/30$) (Brown, 2011). The limit of separability corresponds to the minimum vertical distance between two interfaces required for them to produce distinct seismic reflections within a survey (Brown, 2011). If the vertical distance between two interfaces is between the limits of separability and visibility, their reflections will interfere and cannot be deconvolved; i.e. they produce tuned reflection packages (Brown, 2011). Interfaces separated by vertical distances less than the limit of visibility will be indistinguishable from noise (Brown, 2011). Our calculations indicate the





limits of separability and visibility at the top of the interval of interest, within the Early Cretaceous Barrow Group, are ~19–25 m and ~2–3 m, respectively. Towards the base of the 3D seismic surveys at ~5.9–6.0 s TWT, the limits of separability and visibility decrease to ~80–320 m and ~11–43 m, respectively. The lateral resolution of the data similarly decreases with depth, from 10–12 m to ~31–62 m.

**3.3 Errors**

Here we carefully consider the errors associated with our quantitative analysis of VZD geometry. For example, synthetic seismic forward modelling indicates dyke-related VZD width is dependent on data quality and resolution, and thus likely does not equal dyke thickness (Eide et al., 2018). Data quality and resolution, in turn, is influenced by a range of geophysical (e.g., acquisition and processing parameters) and geological (e.g., faults may locally inhibit imaging) factors. The different acquisition and processing histories of the seismic surveys we use, coupled with spatial variations in the geology of the study area, therefore makes it challenging to assess the likely errors associated with our measurements of VZD width and spacing; i.e. we cannot easily determine how closely the VZD geometry reflects the thickness and spacing of the structures they correspond to (e.g., Fig. 5). The local strike and dip of VZDs may also potentially differ from that of their corresponding structure(s) (e.g., Fig. 5), although we consider these variations to be negligible given their high length-to-width and height-to-width aspect ratios. Because we do not know how seismic velocity varies laterally away from areas of borehole control, we do not depth-convert the seismic reflection data, instead presenting measurements in time (milliseconds TWT) rather than depth (in metres). However, to help geoscientists working with field data and to provide an overall sense of scale, we provide approximate depth-converted value (in metres) for each measurement in time but we cannot ascertain the accuracy of these conversions. Overall, in an attempt to account for and visually communicate the potential errors described above, as well as those introduced by human bias during interpretation and measurement, we conservatively consider that each quantitative parameter could have an arbitrary error of either: (i) ±10% if the property analysed is measured in time (e.g., VZD upper tip depth); or (ii) ±50 m if distances (e.g., VZD length, width, and spacing) are measured in plan-view.

**4 Results**

**4.1 Vertical zones of disruption (VZD)**

**4.1.1 Seismic expression**

We mapped 26 (A-Z) major VZDs, three of which comprise closely overlapping but apparently physically unconnected sections (i.e. VZDs B.1-B.2, G.1-G.6, and H.1-H.2; Fig. 4). The VZDs are broadly planar and dip at ≥80° (e.g., Figs 5-7). Where data quality is high, stratigraphic reflections within the VZDs are deflected upwards, displaying chevron-like geometries, and typically have lower amplitudes relative to their regional attitude (e.g., Figs 6a and b). In places, the VZDs cross-cut igneous sill-related reflections, which are similarly deflected upwards (e.g., Fig. 6b). Where data quality is lower,





the VZDs are subtle and typically only marked by a reduction in amplitude and/or minor geometrical distortion of the stratigraphic reflections they cross-cut (e.g., Figs 6d-f and 7). On some 2D and 3D seismic sections, particularly where data quality is poor and tectonic faults inhibit imaging, we could not recognise VZDs in locations where we predicted them to

occur based on their along-strike projection (e.g., Fig. 7c). Conversely, we identified some additional VZDs on individual 2D seismic lines but could not map these on neighbouring sections located as little as 5 km along-strike (e.g., Fig. 6f); in these cases it was difficult to determine if the VZDs truly terminated along-strike, or whether they were simply not imaged on adjacent lines. Where VZDs cross-cut pre-existing fluvial channels or linear structures within the Mungaroo Formation, there is no resolved vertical or lateral offset of these potential host rock strain markers (e.g., Figs 5 and 8).

### 4.1.2 Borehole expression


The deviated Chester-1 ST1 well intersects VZD H.1 at a depth of ~4.7–5.0 km (Figs 9a-c) (Childs et al., 2013). Where they intersect, the borehole has an inclination of 18° (from vertical), whereas VZD H.1 is ~130±50 m wide, strikes ~003°, and dips at ≳80° W (Figs 9a-c). Cuttings and well-log data reveal the sampled section of VZD H.1 comprises a siliciclastic sedimentary sequence that contains a 48 m thick interval of altered basalt between 4.911–4.959 km (Fig. 9d) (Childs et al.,

2013). Compared to the encasing siliciclastic rock, the altered basalt has a low gamma ray (down to ~6 API) and neutron porosity (down to ~7 pu) signature, but relatively high density (up to ~2.9 g cm$^3$), resistivity (~6200 ohm m), and acoustic slowness (~>90 ms ft) values (Fig. 9d) (Childs et al., 2013). An intra-Mungaroo seismic reflection coincident with the identified basalt has a negative polarity and locally displays a moderate amplitude (Figs 9a-c). Where VZDs H.1 and H.2 cross-cut the intra-Mungaroo reflection, its amplitude is locally reduced (Fig. 9c).

### 4.1.3 Geometry

In plan-view, the VZDs are linear, ranging in length ($L$) from ~4–171 km and with tip-to-tip strikes ($S$) between 353° and 021° (Figs 4 and 10b; Table 1). Overall, the VZDs have a mean $S$ of 008° and broadly display a westwards progression from ~NNE-SSW striking to ~NNW-SEE striking (Figs 4 and 10b). Only the ~N-S striking (002°) VZD B intersects other VZD traces (i.e. VZDs C and D; Fig. 4); the resolution of the data is insufficient to determine whether the VZDs merge at these

intersections, or if one cross-cuts and potentially offsets the other. Depending on their form between the Thebe and HEX03A datasets, where tectonic faulting inhibits their imaging on the intervening 2D seismic lines, VZDs S-Y may also intersect or connect (Figs 4 and 7). Along most (94%) of the mapped VZDs, minor but abrupt changes in strike allow us to sub-divide them into numerous connected segments (Figs 4, 5, and 9c). Across the mapped VZDs, we recognise 280 discrete segments (e.g., Dyke H.1 comprises 26 segments), which have strikes ($s$) between 350° and 044°, and lengths ($l$) of 0.4±0.05 to

33.1±0.05 km (Figs 4 and 10b; Supplementary Table S3). Both $L$ and $l$ display a relatively good-fit with log-normal and negative exponential distributions, and poorer fits to normal and power-law distributions (Fig. 10c).

The depth of VZD upper tips can be mapped relatively accurately within 3D seismic surveys, although convergence of overlying graben-bounding normal faults can locally inhibit their imaging (e.g., Figs 6 and 7). Within the Glencoe,



Chandon, Centaur, and Colombard 3D surveys, the upper tips of VZDs occur between 3.4±0.34 s TWT and ~4.5±0.45 s
TWT (~3.5–5.8 km) (Figs 6a-e and 11a; Supplementary Table S4); the upper tip depths of these VZDs have a combined
geometric mean of 3.7±0.37 s TWT (~4.1 km) and a standard deviation of 0.2±0.02 s TWT. The upper tips of VZDs imaged
within the Thebe and HEX03A 3D seismic surveys, which lie in the western part of the study area, occur at ~3±0.3 s TWT
(~2.9 km) (e.g., Fig. 7). Regardless of their precise depth, VZD upper tips across the study area are consistently located ≳1 s
TWT beneath the near Base Cretaceous unconformity (e.g., Figs 6 and 7). The expression of all VZDs, at some point along
their length, continues below ~5 s TWT (~7 km), where they either appear to terminate or extend beneath the survey limit
(e.g., Figs 6 and 7). Although we cannot determine whether the observed lower tips of the VZDs truly mark the base of the
structure they correspond to, our data suggests VZD heights are typically ≳1.5 s TWT (≳3.5 km) and potentially ≳3 s TWT
(≳9 km) in places (e.g., Figs 6 and 7). Only on a few seismic sections, where data quality is high, do we observe undisturbed
reflections directly beneath a VZD, allowing us to constrain its height (e.g., Fig. 6c). For example, the depth to the base of
VZD E appears to decrease northwards from ≳5.8±0.55 s TWT to ~4.4±0.45 s TWT (~8.5–5.6 km) (e.g., Figs 6a-c).

The width ($w$) of the VZDs ranges from 68±50 m to 335±50 m (Fig. 11b; Supplementary Table S4). In places, $w$
could not be confidently measured because other structures (e.g., tectonic faults) locally inhibit VZD imaging. We note $w$
varies between different 3D seismic datasets, each of which had different acquisition geometries, processing histories, and
data quality (Fig. 11b). Regardless of these relatively short-wavelength changes in $w$, there is an apparent overall reduction
in w northwards marked by a weakly negative trend-line for the combined dataset (Fig. 11b).

Spacing ($h$) between individual VZDs is variable across the measured transects but broadly increases northwards
and is best either described by a log-normal or negative-exponential (Figs 11c and d; Supplementary Table S4). For example,
$h$ between VZDs D-E and G-H increases northwards from ~2.77±0.05 km to 4.90±0.05 km and ~6.17±0.05 km to
11.60±0.05 km, respectively (Fig. 11c). A prominent exception to this spatial trend in $h$, is the northwards reduction in h
between VZDs C-D from ~6.80±0.05 to 3.29±0.05 km (Fig. 11C). For part of the lengths of VZDs B-D and B-E, $h$ also
decreases northwards, although this is a function of the different orientation of VZD B relative to the other two VZDs (Figs 4
and 11c). Between physically unconnected VZD sections (e.g., G.1-G.6), h is ≲2.01 km, with a minimum of ~0.31±0.05 km
(Fig. 11c). Superimposed onto the large-scale variations in $h$ are localised increases in h (Fig. 11c). The boundaries of these
localised increases in $h$ typically coincide with zones where physically unconnected VZD parts terminate, or where VZDs
contain a short segment with a markedly different trend to its neighbouring segments (Figs 4 and 11c). There is a good-fit
between $h$ and log-normal and negative exponential distributions, but the fit of $h$ to normal and power-law distributions is
poorer (Fig. 11d).

## 4.2 Structures associated with VZDs

Overlying and parallel to most VZDs are either one or two, large (up to ~170 km long) normal fault systems, which dip
towards and typically converge on the uppers tips of the VZDs (e.g., Figs 6, 7, and 12). In plan-view, these broadly linear
normal fault systems extend along much of the length of the underlying VZD (Fig. 12). The normal fault systems commonly





comprise multiple low-throw (≲0.1±0.02 s TWT; ≲160 m) faults that are up to ~24 km long (Figs 6, 7, and 12). Only four VZDs (K, L, M, and O), as well as southern portions of VZDs H, G, and U, are not overlain by normal fault systems (Fig. 12); this apparent absence of faults may be real, or could be because much larger tectonic normal faults inhibiting imaging of

smaller, VZD-related structures. Individual faults within the larger systems extend upwards from the tops of VZDs and terminate within Late Jurassic-to-Early Cretaceous strata, bounding graben of half-graben (e.g., Figs 6 and 7). Laterally restricted antithetic and synthetic normal faults occur within these graben and half-graben (e.g., Figs 6A and 7b-c). The youngest stratigraphic horizon offset by the majority of VZD-related normal faults is the near Base Cretaceous unconformity (~148 Ma), although some faults appear to extend upwards into and link with a polygonal fault tier within the Barrow Group

(e.g., Figs 6 and 7).

Sub-circular depressions occur within the graben and half-graben overlying the VZDs (Figs 12 and 13). These depressions are located at the near Base Cretaceous unconformity or at slightly deeper stratigraphic levels within the Dingo Claystone (e.g., Figs 6a, e, 7b, and 13). The depressions are up to ~0.5 km wide, ≲50 ms TWT (≲80 m) deep and infilled by overlying strata (e.g., Figs 6a, e, 7b, 12, and 13). Sub-vertical pipes, within which seismic reflections are displaced

downwards relative to their regional trend, underlie each depression (e.g., Figs 6a, e, 7b, and 13). These pipes extend down to the underlying VZD tip or terminate within the Mungaroo Formation above the corresponding VZD (e.g., Figs 6a, e, 7b, and 13).

## 5 Interpretation

The VZDs define a 'swarm' of up to ~171 km long, relatively thin (<335±50 m wide), sub-vertical, sub-planar zones (Figs 4-

7). These zones cross-cut and disrupt the continuity and amplitude of stratigraphic reflections within the Mungaroo Formation and likely older sedimentary sequences (Figs 5-7). We are confident the VZDs are not the manifestation of geophysical artefacts, but are instead real geological features given they: (i) occur across multiple 2D and 3D seismic datasets with different acquisition and processing histories (e.g., Figs 4-7; Supplementary Table S1); and (ii) are oblique to the inline and crossline directions of the 3D seismic surveys, and thus do not represent an acquisition footprint (Fig. 4;

Supplementary Table S1). Where similar VZDs have been recognised in other seismic reflection datasets, they have been shown to correlate with either the presence of fluid escape conduits (e.g., Cartwright and Santamarina, 2015; Jamtveit et al., 2004; Moss and Cartwright, 2010), strike-slip faults (Lemiszki and Brown, 1988; Schweig III et al., 1992; Harding et al., 1985; Harding, 1985), or igneous dykes (e.g., Wall et al., 2010; Ardakani et al., 2017; Kirton and Donato, 1985; Holford et al., 2017; Minakov et al., 2018; Plazibat et al., 2019).

We discount fluid escape as an origin for our VZDs because these events produce laterally restricted, pipe-like conduits that are geometrically very different to the elongate planar features we observe here (Fig. 4) (e.g., Cartwright and Santamarina, 2015; Jamtveit et al., 2004; Moss and Cartwright, 2010). We also demonstrate that fluvial channels and linear structures within the Mungaroo Formation are not vertically or laterally offset by cross-cutting VZDs (e.g., Figs 5 and 8),

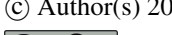



indicating there is no evidence for strike- or dip-slip motion across the latter (cf. Harding, 1985). Plate reconstructions for the
time of break-up between Greater India and Australia in the Late Jurassic-to-Early Cretaceous, informed by the orientation
of tectonic normal faults, seafloor spreading anomalies, and the Cape Range Fracture Zone, further suggest rifting was
margin-parallel and thus unlikely to involve significant ~N-trending, strike-slip faulting (e.g., Heine and Müller, 2005). We
therefore consider it unlikely that the VZDs are faults.

We interpret the VZDs as igneous dykes because: (i) their seismic expression appears similar to dykes in other real
and synthetic seismic datasets (cf. Figs 2, 6, and 7) (e.g., Wall et al., 2010; Ardakani et al., 2017; Kirton and Donato, 1985;
Holford et al., 2017; Minakov et al., 2018; Plazibat et al., 2019; Eide et al., 2018); and (ii) the geometry of individual VZDs,
as well as that of the array they comprise, are akin to the morphology of dyke swarms exposed at Earth's surface (cf. Figs
1A-B and 4) (e.g., Jowitt et al., 2014; Ernst et al., 2001; Halls, 1982). The ~48 m thick basalt interval intersected by the
Chester-1 ST1 well, which occurs within VZD H.1, may further support our interpretation that the VZDs correspond to
igneous dykes (Fig. 9). However, to attribute the recovered basalt cuttings to a dyke, we first need to assess whether the well
could instead have penetrated a lava flow or sill. Based on an interval velocity of ~4.7±0.5 km s$^{-1}$ and a dominant frequency
of ~20 Hz around the intersected basalt, we calculate that the limits of separability and visibility are locally ~59±6 m and
~8±1 m, respectively. Given these limits of separability and visibility, coupled with the higher density and seismic velocity
of the basalt compared to the surrounding sedimentary rocks (Fig. 9d), a ~48 m thick lava flow or sill should be seismically
expressed as a high-amplitude, positive polarity, tuned reflection package (e.g., Eide et al., 2018; Rabbel et al., 2018). Yet
the intra-Mungaroo seismic reflection coincident with the basalt in Chester-1 ST1 has a negative polarity and is of moderate
amplitude (Figs 9A and B). These observations suggest the basalt intersected by Chester-1 ST1 does not come from a lava
flow or sill, but instead supports our interpretation that the coincident VZD H.1, and likely other VZDs, are igneous dykes.
We concede that limitations in seismic and well data resolution mean we still cannot determine whether individual VZDs
correspond to a single dyke or multiple, closely spaced dykes (e.g., Fig. 5).

Our interpretation that the VZDs correspond to igneous dykes raises the question as to whether the observed
overlying normal fault systems and pipes, which converge on the inferred dykes, were genetically related to magmatism
(e.g., Figs 4, 6, 7, 12, and 13). For example, normal fault systems and sub-circular depressions similar to those we describe
have been observed above dykes on Earth, other planetary bodies, and in physical and numerical models (e.g., Wyrick et al.,
2004; Wilson and Head, 2002; Trippanera et al., 2015b; Trippanera et al., 2015a; Rubin and Pollard, 1988; Pollard et al.,
1983; Rubin, 1992; Hardy, 2016; Wyrick and Smart, 2009; Okubo and Martel, 1998). Numerical and analytical models
suggest normal faulting above intruding and widening dykes is driven by the concentration of tensile stress at the dykes
upper tip and at the contemporaneous surface (e.g., Rubin and Pollard, 1988; Pollard et al., 1983; Rubin, 1992). Shear failure
within this local dyke-induced stress field produces graben- or half graben-bounding, dyke-parallel normal faults that dip
towards and converge on the dykes upper tip (e.g., Rubin and Pollard, 1988; Pollard et al., 1983; Rubin, 1992; Trippanera et
al., 2015b); these faults are termed 'dyke-induced normal faults'. Dyke intrusion and widening can also locally produce
cavities through the accumulation and release of magmatic volatiles at its upper tip, or the heating and escape of pore fluids



in the immediately overlying host rock (e.g., Wyrick et al., 2004; Wilson and Head, 2002; Mège et al., 2003). Collapse of these cavities produces overlying pipe-like zones of subsidence expressed at the contemporaneous surface as sub-circular

depressions called 'pit craters' (e.g., Wyrick et al., 2004; Wilson and Head, 2002; Mège et al., 2003). Due to their spatial coincidence with underlying dykes, and given their geometrical similarities to supra-dyke structures observed elsewhere, we suggest the faults and depressions described here are dyke-induced normal fault systems and pit craters (Figs 5, 7, 12, and 13) (e.g., Wyrick et al., 2004; Wilson and Head, 2002; Trippanera et al., 2015b; Trippanera et al., 2015a; Rubin and Pollard, 1988; Pollard et al., 1983; Rubin, 1992; Hardy, 2016; Wyrick and Smart, 2009; Okubo and Martel, 1998).

## 6 Discussion

### 6.1 Timing of dyke emplacement

Radiometric dates are unavailable to constrain the emplacement age of the studied dykes, so we have to apply seismic-stratigraphic techniques. Each dyke intrudes and terminates within the Mungaroo Formation, indicating their emplacement occurred during or after the Triassic (e.g., Figs 6, and 7). The dykes also cross-cut and thus post-date sills intruded within the

Triassic Mungaroo Formation (e.g., Figs 6b, c, and f). Although we have no constraints on the age of these sills cross-cut by the dykes, it is likely they were emplaced during a regional phase of Late Jurassic-to-Early Cretaceous magmatism (e.g., Symonds et al., 1998; Rohrman, 2013; Magee et al., 2017; Magee et al., 2013b; Magee et al., 2013a). Onlap of overlying strata onto intrusion-induced forced folds suggest sill emplacement elsewhere in the North Carnarvon Basin may have begun in the Kimmeridgian (Magee et al., 2017; Magee et al., 2013a).

370        The near Base Cretaceous unconformity (~148 Ma) is the youngest stratigraphic horizon deformed by most of the interpreted dyke-induced normal fault systems and pit craters (e.g., Figs 6 and 7). Where dyke-induced normal fault systems and pit craters are observed elsewhere on Earth or other planetary bodies, they deform the surface, contemporaneous with dyke intrusion (e.g., Wyrick et al., 2004; Wilson and Head, 2002; Trippanera et al., 2015b; Trippanera et al., 2015a; Rubin and Pollard, 1988; Pollard et al., 1983; Rubin, 1992; Hardy, 2016; Wyrick and Smart, 2009; Okubo and Martel, 1998). Our

seismic-stratigraphic observations therefore suggest the near Base Cretaceous unconformity (~148 Ma) likely marked the palaeosurface during dyking, indicating emplacement principally occurred during or after its development, but ceased before the overlying Barrow Group was deposited. Some pit craters terminate within rather than at the top of the Dingo Claystone (e.g., Fig. 13), suggesting dyking may have initiated in the Late Jurassic before the near Base Cretaceous unconformity formed at ~148 Myr. The apparent extension of some dyke-induced normal faults into the ~146.7–138.2 Ma Barrow Group,

which is located above the near Base Cretaceous unconformity, may be indicative of renewed, post-Valanginian dyking (Figs 6d, e, and 7a-c). An alternative suggestion is that the upward extension of the dyke-induced normal faults into the Barrow Group simply reflects fault reactivation and/or dip-linkage during later polygonal fault formation (i.e. these fault extensions are unrelated to dyking). Such reactivation or dip-linkage of the dyke-induced normal faults is supported by the: (i) reduced dip of many dyke-induced faults segments above the near Base Cretaceous unconformity (e.g., Figs 6d, e, and 7a-





b); and (ii) similar extension of some tectonic normal faults above the near Base Cretaceous unconformity, occasionally to just below the seabed. Overall, we propose all dykes were likely intruded during a short period in the latest Jurassic, probably during the Tithonian (~152–147 Ma), before the onset of Barrow Group deposition at ~146.7 Ma (Reeve et al., 2016); we name this newly discovered suite of igneous dykes the Exmouth Dyke Swarm.

## 6.2 Dyke swarm structure

To understand the kinematics and mechanics governing dyke swarm emplacement, we typically rely on measuring the geometrical properties (e.g., length, width, and spacing) of dykes exposed at the Earth's surface (e.g., Paquet et al., 2007; Jolly and Sanderson, 1995; Gudmundsson, 1983). A potential problem with these analyses is that we can only measure the surface, principally 2D expression of dykes and dyke swarms, which may not equal their true 3D geometry. For example, seismic reflection data from offshore southern Norway reveal the width of an imaged dyke swarm increases with depth,

implying the dimensions of dyke swarms measured at the surface depend partly on erosion level and may therefore not capture the true swarm geometry (Phillips et al., 2018). Seismic reflection data thus provide a unique opportunity to examine and quantify the 3D structure of a dyke swarm independent of the potential bias introduced by the processes (e.g., erosion) controlling how dyke swarms intersect the surface. Here, we specifically discuss how our measurements of dyke length, thickness, and spacing compare to predicted distributions of these geometrical properties derived from surface- and physical,

numerical, and analytical modelling-based studies.

### 6.2.1 Dyke length

Lengthening of fractures is commonly facilitated by linkage between individual segments (e.g., Mège and Korme, 2004; Schultz, 2000; Cladouhos and Marrett, 1996; Gudmundsson, 1987). The evolution of a fracture population can be unravelled from its length distribution if we can ascertain whether linked or closely spaced fractures should be treated as one or several

structures (e.g., Schultz, 2000; Mège and Korme, 2004); i.e. does the length-frequency distribution of a fracture population change through time in response to linkage modifying the behaviour of the system, or is it scale invariant? Dykes are magma-filled fractures and can broadly be considered to intrude instantaneously and independently formed fractures (i.e. they do not interact), implying the length-frequency distribution of a dyke swarm should preserve the initial configuration of the fracture population (Mège and Korme, 2004). Comparing data from fracture and dyke populations reveal their length-

frequency distributions are both broadly power-law, suggesting mechanical linkage of fractures does not modify system behaviour (e.g., Mège and Korme, 2004; Schultz, 2000; Cladouhos and Marrett, 1996; Gudmundsson, 1987; Paquet et al., 2007). Here we use our data, assuming the dykes are Mode I fractures, to examine whether: (i) measurement of dyke-surface intersections introduces bias to length-frequency distributions; and (ii) dyke segmentation, which may be indicative of non-instantaneous and non-independent fracture growth, also display a power-law length-frequency distribution (cf. Mège and

Korme, 2004).





Cumulative length-frequency plots for all measured dyke lengths (*L*), which comprise connected and/or closely spaced but physically unconnected segments, initially appear to fit a log-normal or negative exponential, rather than a power-law distribution (Fig. 10c) (cf. Paquet et al., 2007; Mège and Korme, 2004). Dyke segment length (*l*) data display similar log-normal and negative exponential distribution characteristics (Fig. 10c). However, power-law distributions can be

fit to *L* values between 20–160 km and *l* values of 5–20 km (Fig. 10c). The population exponents (*C*) for the *L* and *l* datasets are 1.29 and 2.85, respectively, consistent with values derived from the analysis of other fracture and dyke populations (Fig. 10c) (see Mège and Korme, 2004 and references therein). The observed departure of our measured *L* and *l* values from a power-law distribution at small and large length-scales could indicate bias in the data. For example, restrictions in dyke imaging and 2D seismic line spacing may mean: (i) the dykes are likely longer than mapped; (ii) some dykes (e.g., VZDs X

and Z) may be connected along-strike, thereby increasing their length (Figs 4b-d); and (iii) small dykes and/or dyke segments are difficult to recognise or may not be imaged because they occur between 2D seismic lines outside of areas imaged by the 3D surveys. We contend that our data could thus be considered consistent with previous studies in describing dyke length distributions as power-law, indicating processes controlling dyke length (e.g., segmentation) are scale invariant (Mège and Korme, 2004). Furthermore, our results suggest the free-surface intersection of fractures or dykes is, at least

typically, representative of a population's length distribution.

### 6.2.2 Dyke thickness

The thickness of a dyke, or cumulative thickness of a dyke swarm, influences a variety of processes, including eruption rates and crustal extension (e.g., Krumbholz et al., 2014). Statistical analyses of dyke thickness distributions derived from surface-based measurements inform dynamic models of dyke emplacement, shedding light on the processes controlling dyke

thickness (e.g., Krumbholz et al., 2014; Jolly and Sanderson, 1995; Klausen, 2006; Klausen, 2004). Resolving the 3D structure of dyke swarms in seismic reflection data provides a potential opportunity to examine both lateral and vertical variations in dyke thickness distribution. However, synthetic seismic forward models suggest the width of VZDs corresponding to sub-vertical dykes is greater than the true thickness of the dyke itself (Eide et al., 2018). Furthermore, because VZD width is partly controlled by the acquisition and processing properties of the data in which they are imaged in

(e.g., frequency; Eide et al., 2018), evidenced by the marked differences in VZD width between different seismic surveys (Fig. 11c), it is difficult to determine how VZD width and true dyke thickness are related. We show VZD widths measured across multiple 3D seismic surveys gradually decrease northwards (Fig. 11c). Because the northwards decrease in VZD width is consistent across multiple seismic surveys, which each have different acquisition and processing parameters, we suggest this trend marks a similar northwards decrease in true dyke thickness (Fig. 11c). From the Chester-1 ST1 well, which

likely intersects a 48 m long section of basalt, we calculate the dyke has a true thickness of ~18 m, assuming its orientation is parallel to that of the ~130±50 m wide VZD it relates to (Fig. 14). This well data confirms synthetic seismic forward model predictions that dyke-related VZD width is much greater than true dyke thickness (Eide et al., 2018). Further work in





understanding how dykes are expressed in seismic reflection data is required before these data can be used to quantify dyke thickness distributions.

**6.2.3 Dyke spacing**

Plan-view sections through dyke swarms reveal individual dykes are typically regularly spaced, with the spacing (*h*) of radiating swarms increasing away from their focal area (e.g., Bunger et al., 2013; Jolly and Sanderson, 1995; Ernst et al., 1995). Identifying controls on h is fundamental to understanding why dykes occur in swarms and, thus, how they interact with and/or drive crustal extension on Earth and other planetary bodies (Bunger et al., 2013). Analytical predictions suggest

first-generation, laterally propagating dykes will have energetically optimal spacings that are related to dyke height (*H*) and magma source conditions (Bunger et al., 2013). For dykes emanating from a constant pressure magma source (i.e. an infinitely large, compressible reservoir), *h*/*H* is expected to be ≈ 1, whilst those from a constant influx magma source (i.e. a small, incompressible reservoir) will have either a *h*/*H* of ≈ 2.5 or ≈ 0.3 (Bunger et al., 2013). Constraining the relative age of dykes is critical to testing these analytical predictions because second-generation or younger may preferentially intrude

between first-generation dykes, thereby reducing the apparent spacing (Bunger et al., 2013).

Dyke spacing within the Exmouth Dyke Swarm ranges from ~22.4±0.05 km to 0.3±0.05 km, with a geometric mean of ~4.1 km, and broadly increases northwards (Figs 4 and 11d). This northward increase in *h*, coupled with apparent northwards reductions in dyke thickness *and* abundance, implies extension accommodated by the Exmouth Dyke Swarm similarly decreased northwards (Figs 4 and 11d). To test analytical predictions using our measured *h* values, it is first

important to recognise key limitations in our dataset: (i) not all dykes within the swarm may be imaged by the seismic reflection data, suggesting our *h* measurements are likely only maximum values; (ii) *H* is difficult to quantify because a reduction in data quality with depth likely means we cannot accurately pick the lower tips of dykes, some of which may extend beneath the seismic surveys (e.g., Figs 6 and 7); and (iii) it is challenging to ascertain whether all dykes were emplaced simultaneously or not during the latest Jurassic dyking event. Because the dykes are typically >1.5 s TWT tall

(e.g., Figs 6 and 7), we use extrapolated checkshot data to estimate the average *H* is at least ~3.5 km (Supplementary Fig. S3). As a maximum estimate for average *H*, we consider the dykes could extend upwards from the base of the crust, which across the Exmouth Plateau is likely ~20–28 km beneath the present day seabed (e.g., Reeve et al., 2016; Tindale et al., 1998; Stagg et al., 2004; Stagg and Colwell, 1994; Mutter and Larson, 1989). Given the upper dyke tips broadly occur at ~3.7±0.37 s TWT, equivalent to a depth of ~4.1 km, we therefore suggest the maximum average *H* could be up to ~24 km.

Assuming dyking was instantaneous and using the geometric mean for *h* (~4.1 km), we calculate *h*/*H* ≈ 1.17–0.17.

The calculated h/H values of 1.17–0.17 are broadly consistent with and cannot be used to discriminate between the constant pressure (*h*/*H* ≈ 1) and constant influx (*h*/*H* ≈ 0.3) end-member source conditions (Bunger et al., 2013). However, if seismically unresolved dykes are present in the study area, we may expect h to be less than that measured and thus more consistent with *h*/*H* ≈ 0.3, implying the dykes were fed from a constant influx magma source (Bunger et al., 2013).

Alternatively, if we consider dyking was incremental, with later dykes intruding host rock between pre-existing intrusions,





we would expect $h \gtrsim 4.1$ km for the first-generation dykes; this would imply the original maximum $h/H$ ratio could be $\approx 1$. Potential evidence for incremental emplacement of the Exmouth Dyke Swarm includes: (i) the relatively good fit of $h$ to a negative-exponential distribution (Fig. 11d), which suggests $h$ is random and likely results from incorporation of different dyke sets into the data; and (ii) the observation that some pit craters occur within (rather than at the top of) the Dingo

Claystone (e.g., above Dyke F; Fig. 13), suggesting their associated dykes were emplaced before the formation of the near Base Cretaceous unconformity (~148 Ma). For example, if we hypothetically consider VZDs C, F, H, and I were emplaced first, their geometric mean h of 12.4 km implies $h/H \approx 3.54$–0.52, which again could be considered consistent with a constant pressure ($h/H \approx 1$) or constant influx ($h/H \approx 2.5$) source (Bunger et al., 2013). Mapping the occurrence and distribution of pit craters formed before the near Base Cretaceous unconformity may allow us to identify first-generation dykes and thereby

constrain dyke source conditions.

### 6.3 Emplacement of the Exmouth Dyke Swarm

We mapped the Exmouth Dyke Swarm, as well as associated dyke-induced normal faults and pit craters, across a ~40,000 km2 part of the North Carnarvon Basin (Figs 4-7 and 12). Long, linear graben, containing sub-circular depressions, similar to the dyke-induced normal faults and pit craters we identified, occur at the near Base Cretaceous unconformity elsewhere in

the North Carnarvon Basin (e.g., Fig. 15) (Velayatham et al., 2019; Velayatham et al., 2018). The formation of some of these other depressions has been linked to fluid escape following faulting of overpressure strata, and *not* dyking (Velayatham et al., 2018). However, their geometrical similarity to and occurrence at the same structural level as the dyke-induced normal fault systems and pit craters described here suggests they could be the palaeosurface expression of the Exmouth Dyke Swarm (cf. Figs 12 and 15) (see also Velayatham et al., 2019). This potential distribution of dykes (except for VZD K),

dyke-induced normal fault systems, and pit craters across the North Carnarvon Basin appears to describe a giant radial dyke swarm (cf. Figs 1c and 15c) (cf. Ernst, 2014; Ernst et al., 2001; Ernst et al., 1995; Halls and Fahrig, 1987). Projecting the inferred dykes to a common focal area, which is located on the Cuvier Margin, suggests the Exmouth Dyke Swarm could be >500 km long and distributed around a ~039° (perhaps up to ~054°) arc (Fig. 15c). To unravel the origin of the Exmouth Dyke Swarm, we first discuss evidence for magma propagation direction and syn-emplacement stress conditions.

### 6.3.1 Dyke propagation direction

The radiating form of the Exmouth Dyke Swarm suggests individual dykes may have been sourced and thus flowed laterally northwards from the northern sector of the Cuvier Margin (Fig. 15c) (see also Velayatham et al., 2019). Lateral propagation of the dykes to the north is supported by the: (i) maintenance of dyke upper tip depths (Figs 6, 7, and 11a), consistent with the expectation that horizontally emplaced dykes have fixed upper and lower tip positions (e.g., Townsend et al., 2017); (ii)

subtle northwards decrease in VZD width (Fig. 11b), which we suggest reflects thinning of dykes, perhaps towards their laterally propagating tip (e.g., Healy et al., 2018); and (iii) minor but abrupt changes in the strike of connected dyke


segments (Figs 4 and 5), which are reminiscent of the kinked geometry attained by the Bárðarbunga-Holuhraun dyke during its incremental, lateral propagation (Woods et al., 2019; Sigmundsson et al., 2015).

### 6.3.2 Palaeostress conditions during dyke emplacement

The orientation and structure of dykes and dyke swarms is commonly used to reconstruct syn-emplacement stress and magma conditions (e.g., Odé, 1957; Hou et al., 2010; Grosfils and Head, 1994; Lahiri et al., 2019; Jolly and Sanderson, 1997; Jolly and Sanderson, 1995). Deriving these overarching controls on dyke emplacement assumes that dykes preferentially develop orthogonal to $\sigma_3$ within the $\sigma_1$-$\sigma_2$ plane (e.g., Anderson, 1951). Although the orientation of dykes and dyke segments studied here is variable, they are broadly N- to NE-trending and sub-vertical (~80–90°), suggesting an
average syn-emplacement $\sigma_3$ currently oriented 100/00° (Fig. 16). Mutually orthogonal to the calculated $\sigma_3$ on a lower-hemisphere, equal area stereographic projection are two axes, at 010/00° and 280/90° respectively, which can be ascribed to $\sigma_1$ or $\sigma_2$ depending on their proximity to the cluster of measured dyke poles (e.g., Jolly and Sanderson, 1997; Lahiri et al., 2019). Specifically, the angle measured along the $\sigma_1$-$\sigma_3$ plane between the cluster of dykes and $\sigma_1$ (i.e. $\theta2$) will be greater than that measured along the $\sigma_2$-$\sigma_3$ plane between the data and $\sigma_2$ (i.e. $\theta1$; Fig. 16) (e.g., Jolly and Sanderson, 1997; Lahiri et
al., 2019). Our data thus suggests that during dyking, the overarching stress field in the study area was extensional with a vertical $\sigma_1$ (000/90°) and horizontal, N-trending $\sigma_2$ (010/00°) (Fig. 16). The syn-emplacement, ~W-trending, horizontal $\sigma_3$ axis we define is comparable to suggested W- to NW-trending extension directions, estimated from tectonic fault orientations and seafloor spreading patterns, for Late Jurassic-to-Early Cretaceous rifting and break-up offshore NW Australia (e.g., Hopper et al., 1992; Heine and Müller, 2005; Driscoll and Karner, 1998). Where NW-trending dykes may
dominate to the west of the study area (Fig. 15c) (Velayatham et al., 2018), we anticipate the horizontal principal stress axes ($\sigma_2$ and $\sigma_3$) were oriented NW-SE and NE-SW, respectively, whilst $\sigma1$ remained vertical.

### 6.3.3 Tectono-magmatic setting and source of the Exmouth Dyke Swarm

Magmatism across the North Carnarvon Basin has been attributed to decompression melting during rifting (Karner and Driscoll, 1999), coupled rifting and small-scale convective partial melting (e.g., Hopper et al., 1992; Mutter et al., 1988;
Mihut and Müller, 1998), and/or mantle plume activity (e.g., Rohrman, 2015, 2013; Müller et al., 2002; Mihut and Müller, 1998). We show emplacement of the Exmouth Dyke Swarm occurred during the latest Jurassic (~152–147 Ma), after intrusion of extensive sill-complexes (e.g., Figs 6 and 7). Individual dykes likely propagated laterally away from a source focal area, which we infer was located on the Cuvier Margin, SSE of the study area (Fig. 15c). Dyking and earlier sill emplacement thus predated the main phase of igneous activity recorded across the North Carnarvon Basin, which was
associated with formation of the ~136–130 Ma continent-ocean transition zones bordering the Gascoyne and Cuvier margins, and ultimately continental break-up in the Hauterivian (e.g., Rey et al., 2008; Symonds et al., 1998; Mihut and Müller, 1998; Reeve et al., 2019; Direen et al., 2007; Robb et al., 2005). Seismic reflection data also reveal there was little upper crustal normal faulting or rifting across the Exmouth Plateau in the Late Jurassic, immediately prior to and during dyking (e.g.,





Driscoll and Karner, 1998; Bilal et al., 2018). These age relationships suggest the Exmouth Dyke Swarm and earlier sills
were likely not associated with rift-related melting, which appears to have initiated in the Early Cretaceous (cf. Mihut and
Müller, 1998; Karner and Driscoll, 1999; Hopper et al., 1992; Mutter et al., 1988). Instead, the large extent and radial
disposition of the Exmouth Dyke Swarm suggests it may have been sourced from either a regional, thermal mantle anomaly
(e.g., a plume or small-scale convection cell) or a large volcanic system (e.g., Ernst and Buchan, 1997; Ernst et al., 1995;
Odé, 1957; Speight et al., 1982).

550       Any process invoked to explain the origin of a thermal mantle anomaly in the Late Jurassic, and potentially into the
Early Cretaceous, needs to account for (e.g., Rohrman, 2015, 2013; Müller et al., 2002; Mihut and Müller, 1998; Hopper et
al., 1992; Mutter et al., 1988): (i) the latest Jurassic distribution of magmatism across the Gascoyne and Cuvier margins; and
(ii) denudation patterns and formation of contemporaneous regional unconformities (e.g., the near Base Cretaceous
unconformity). Two possible mantle plume sites on the Cuvier Margin have previously been proposed, with one located on
the Bernier Platform, initiating at ~136 Ma, and the other active on the conjugate to the Cuvier Margin near the current Cape
Range Fracture Zone between ~165–136 Ma (e.g., Fig. 17a) (cf. Rohrman, 2015; Müller et al., 2002; Mihut and Müller,
1998). Mantle plume activity has previously been discounted as a viable source for Late Jurassic-to-Early Cretaceous
magmatism because no clear hotspot tracks have been identified (e.g., Müller et al., 2002), although Rohrman (2015) argued
the Quokka Rise and Zenith Plateau are part of such a track (Fig. 17a). An alternative interpretation to a mantle plume source
is that melting reflects small-scale mantle convection instigated by juxtaposition of thick and thin lithosphere across a
transform margin (e.g., the Cape Range Fracture Zone) (e.g., Müller et al., 2002; Mutter et al., 1988). Because the formation
of transform margins along the NW Australian Shelf occurred during break-up of Greater India and Australia in the Early
Cretaceous (~136–130 Ma), coincident with the age of the proposed Bernier Platform mantle plume, it seems unlikely these
processes could have generated the latest Jurassic Exmouth Dyke Swarm (cf. Müller et al., 2002; Mihut and Müller, 1998).
The interpreted age and distribution of the Exmouth Dyke Swarm thus fits best with the mantle plume model proposed by
Rohrman (2015).

Within the framework of the mantle plume model proposed by Rohrman (2015), melting is expected to have
initiated ~165 Myr ago, leading to emplacement of a mafic-to-ultramafic, high-velocity magmatic body near the Moho and
formation of the Callovian unconformity during associated uplift (i.e. vertical $\sigma_1$; Fig. 18a). This high-velocity magmatic
body likely fed the Late Jurassic sill-complex prior to emplacement of the Exmouth Dyke Swarm (Fig. 18a) (e.g., Rohrman,
2013; Magee et al., 2017; Magee et al., 2013a; Symonds et al., 1998). We suggest that emplacement of this sill-complex
occurred as plume activity waned and uplift ceased, causing the regional stress to relax such that the vertical principal stress
axis became $\sigma_3$ and basin subsidence initiated (Fig. 18b); this change in stress orientation could explain why the ascent of
buoyant magma from the high velocity body formed a sill-complex rather than a vertical dyke swarm. Layering in the
sedimentary basins may also have favoured sill emplacement (Fig. 18b) (see Magee et al., 2016b and references therein).
The apparent transition from sill-complex formation to intrusion of the Exmouth Dyke Swarm in the latest Jurassic marks an
abrupt change in emplacement conditions. To generate the Exmouth Dyke Swarm, which broadly coincided with a phase of





uplift and denudation (i.e. formation of the Base Cretaceous unconformity), we show $\sigma_1$ had become vertical and $\sigma_3$ was circumferential to the swarms focal area (Figs 16a, 18c, and d). We suggest these conditions, which favoured dyking rather than sill-complex emplacement, could have been instigated by a renewed influx of plume material, with the swarm fed either: (i) directly from a thermal mantle anomaly (Fig. 18c); or (ii) via a large intrusive centre located at the southern boundary of the Exmouth Sub-basin, which manifests as a sub-circular (~20 km diameter), positive magnetic anomaly and a zone of disturbance in seismic reflection data (e.g., Figs 17 and 18d) (Müller et al., 2002). Cessation of plume activity immediately after dyking, following removal or reduction of the thermal anomaly, may explain the rapid subsidence (i.e. <0.24 mm yr$^{-1}$) required to accommodate the Late Jurassic-to-Early Cretaceous Barrow Group (cf. Reeve et al., 2016). Overall, our data seemingly support the presence of a mantle plume offshore NW Australia during the Late Jurassic-to-Early Cretaceous (e.g., Rohrman, 2015, 2013; Müller et al., 2002). However, it remains uncertain whether igneous activity coincident with Hauterivian break-up was also related to the presence of a mantle plume or not.

## 6.4 Implications and future studies

Giant dyke swarms are recognised worldwide onshore (e.g., Ernst and Youbi, 2017; Ernst, 2014; Bryan and Ernst, 2008; Coffin and Eldholm, 2005; Coffin and Eldholm, 1994; Bryan et al., 2010; Hou et al., 2010; Halls and Fahrig, 1987; Halls, 1982; Ernst and Baragar, 1992). Projection of these onshore dyke swarms and the known importance of dyking to break-up and formation of magma-rich margins suggests dyke swarms should also be prevalent on offshore continental shelves (see Magee et al., 2019 and references therein). Our work extends a growing consensus that vertical dykes can be recognised in seismic reflection data imaging continental margins (e.g., Wall et al., 2010; Bosworth et al., 2015; Malehmir et al., 2018; Holford et al., 2017; Kirton and Donato, 1985; Ardakani et al., 2017; Jaunich, 1983; Plazibat et al., 2019). Key criteria for defining vertical dykes in seismic reflection data include: (i) identification of thin, long, tall, typically sub-vertical zones of disturbance within otherwise sub-parallel reflections defining the host rock (e.g., Figs 6 and 7) (e.g., Eide et al., 2018; Minakov et al., 2018; Wall et al., 2010); (ii) lack of lateral or vertical offset of host rock strata, best revealed by mapping piercing points (e.g., fluvial channels, pre-existing structures) across inferred dyke-like features (e.g., Figs 5 and 8), which suggests the features are not strike-slip or steeply dipping normal faults; and (iii) potential association with overlying pit craters or dyke-induced normal faults, which are likely easier to resolve and map in seismic reflection data compared to dykes (e.g., Figs 6, 7, 12 and 13). By increasing our collective awareness of how these criteria can be used to identify dykes in seismic reflection data, we expect more dyke swarms will be revealed across continental margins worldwide. Recognition of dyke swarms within seismic reflection data will help us produce better physical models of the subsurface, aiding our understanding of a margins thermal history, and fluid and/or gas plumbing systems of sedimentary basins.

We also demonstrate that mapping dykes, dyke-induced normal faults, and pit craters across vast areas using seismic reflection data provides unprecedented opportunities to resolve and quantify their natural structure in 3D (e.g., Figs 4-13). Future work should focus on: (i) unravelling the geophysical expression of dykes, such that additional and more accurate quantitative data (e.g., dyke thickness) can be recovered; (ii) deciphering the kinematic history of dyke-induced





normal faults, which we may expect should relate to and thus inform dyke structure and emplacement dynamics; (iii) quantifying the geometrical relationship between pit craters and the dyke intrusions driving their formation; and (iv) determining whether dyke-induced normal faults and pit craters can be used to constrain the temporal evolution of a dyke swarm. These four initiatives will provide new insights into and allow us to test hypotheses concerning the 3D structure and
growth of dyke swarms, and their associated structures. We envisage that these findings will improve how we can invert the surface expression of active or ancient dyke swarms, i.e. dyke-induced normal faults and pit craters exposed at the surface of Earth or other planetary bodies, to recover more information on their otherwise inaccessible subsurface structure and the processes that formed them.

## 7 Conclusions

Dyke swarms are ubiquitous on Earth and other planetary bodies. Yet we know little of the 3D structure of dyke swarms because the pseudo-2D nature of planetary surfaces means we can typically only access their plan-view morphology, and then only at the given erosion level. Here we use a suite of seismic reflection datasets from the Exmouth Plateau offshore NW Australia, to map 26, latest Jurassic (~152–147 Ma) dykes in 3D across ~40,000 km$^2$; we name this the Exmouth Dyke Swarm. The mapped dykes correspond to ~N- to NE-trending, vertical zones of disturbance within the seismic reflection
data that are can be up to 171 km long, ≲355 m wide, likely ≳9 km high, and can be sub-divided into smaller segments with subtly different orientations. Directly above the dykes are a series of graben-bounding normal fault systems, which dip towards and converge upper dyke tips, and sub-vertical pipe-like features; we interpret these structures as dyke-induced normal faults and pit craters. Our quantitative analyses reveal dyke length broadly follows a power-law distribution consistent with previous studies, whilst dyke spacing conforms to a negative-exponential distribution, which we attribute to
sampling of different dyke generations. Across the study area, dyke orientations are consistent with an ENE-trending, horizontal and a vertical minimum and maximum principal stress axes, respectively. However, recognition of possible dyke-induced normal faults and pit craters elsewhere on the Exmouth Plateau suggest dykes are distributed radially across a 39° arc, implying the minimum principal stress axis was circumferential, centred on the Cuvier Margin to the south. This focal area on the Cuvier Margin likely marks the dyke swarm source, which is consistent with evidence the dykes propagated
laterally northwards. Overall, we suggest emplacement of the Exmouth Dyke Swarm related to renewed activity of a mantle plume located on the Cuvier Margin between ~165–136 Ma. Our work demonstrates seismic reflection data can be used to identify vertical dykes across vast areas on continental margins, whilst providing unprecedented into the 3D structure of these natural systems. By defining a series of criteria that can be used to interpret dykes in seismic reflection data, we anticipate future studies will: (i) recognise dyke swarms across continental margins worldwide, providing new insights into
basin evolution (e.g., thermal histories) and controls on fluid flow; (ii) provide more robust constraints on dyke swarm geometry, allowing previous models and hypotheses of their 3D structure to be tested; (iii) reveal how dyke-induced normal faults and pit craters are kinematically linked to dyking; and (iv) demonstrate how dyke swarms may be expressed at the syn-
emplacement surface, meaning we can improve inversions of such surficial features observed on Earth and other planetary bodies to better predict underlying dyke structures.

**Data availability**

The seismic reflection and well data used (see text for details) are publically available through Geoscience Australia at http://www.ga.gov.au/nopims. All measurements presented are within, or based on data within, the compiled Table and Supplementary Tables.

**Author contribution**

CM designed and conducted the research, interpreted the data, and prepared the manuscript. CALJ contributed to data interpretation and manuscript editing.

**Competing interests**

The authors declare that they have no conflict of interest.

**Acknowledgements**

CM is funded by a NERC Independent Research Fellowship (NE/R014086/1). We are grateful to Geoscience Australia for making all data used in this study publically available. Schlumberger are thanked for provision of Petrel seismic interpretation software.

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

**Figure 1: (a) Map of the major dyke swarms on Earth, highlighting their form and age of associated mantle plume sources if relevant (modified from Ernst, 2014; Magee et al., 2019). Dyke swarms shown include: A = 1140 Ma Abitibi swarm, precursor to the 1115–1085 Ma Keweenawan LIP; AA = 30–0 Ma Afar-Arabian swarms; C = 17–0 Ma Columbia River swarms; CAMP = 201 Ma Central Atlantic Magmatic Province swarm; D = 66 Ma Deccan swarm; F = Franklin swarm; G = 779 Ma Gunbarrel swarms;**
**Ga = 799 Ma Gannakouriep swarm; H = 130–90 Ma High Arctic LIP (HALIP) swarm; J = 301 Ma Skagerrak (Jutland) swarms; K = 183 Ma Karoo swarms; M = 2510 Ma Mistassini swarm; Mac = 1267 Ma Mackenzie swarm; Md = 89 Ma Madagascar swarm; Mt = 2480–2450 Ma Matachewan swarm; NAIP = 62–55 Ma North Atlantic Igneous Province swarms (e.g. Mu is the Mull dyke swarm); P-E = ~135–128 Ma Paraná-Etendeka dyke swarms; S = 251 Ma Siberian Traps swarm; U = 2217-2210 Ma Ungava swarm; Y = 370 Ma Yakutsk-Vilyui swarm. Inset: Map of the radial dyke swarm around the Spanish Peaks volcanic centre**



**(redrawn from Odé, 1957). (b-d) Schematic diagrams depicting the relation between mantle sources and parallel/linear (b), radiating (c), and circumferential (d) dyke swarms.**

**Figure 2: (a) Dyke and overlying graben-bounding faults recognised in seismic reflection data from Egypt (modified from Bosworth et al., 2015). Note the dyke corresponds to minor deflections in background stratigraphic reflections. (b) Vertical zone of**
**disturbance within seismic reflection data form the North Sea, where the amplitude of background stratigraphic reflections is relatively diminished and deflected upwards, inferred to be a dyke (Wall et al., 2010). A crater that truncates underlying strata and contains high-amplitude, continuous-to-chaotic reflections is developed above the dyke (Wall et al., 2010).**

**Figure 3: (a) Location map of the southern portion of the North Carnarvon Basin, which spans the Gascoyne Margin and extends**
**onto the Cuvier Margin. Key tectonic elements include: EXP = Exmouth Plateau; DSB = Dampier Sub-basin; BSB = Barrow Sub-basin; ESB = Exmouth Sub-basin; CT = Carnarvon Terrace; MSB = Merlinleigh Sub-basin; and the PS = Peedamullah Shelf. The map also shows the approximate boundary of sill-complexes in the North Carnarvon Basin (modified from Symonds et al., 1998; Holford et al., 2013). (b) Tectono-stratigraphic column for the Exmouth Plateau and Exmouth Sub-basin, which also highlights the relative duration and abundance of Late Jurassic-to-Early Cretaceous magmatism (based on Symonds et al., 1998; Tindale et al.,**
**1998; Longley et al., 2002; Reeve et al., 2016). Undulating lines mark unconformities. (c) Uninterpreted and interpreted seismic section, combining lines AGSO 135/01 and AGSO 110/12, showing the crustal structure of the study area (see Fig. 3a for location). Reflection polarity here, and elsewhere, is defined by a schematic seismic wavelet showing acoustic impedance (A.I.).**

**Figure 4: (a) Location map showing the 2D and 3D seismic surveys and 24 wells used in the study, as well as the plan-view**
**configuration of the 26 vertical zones of disturbance (VZDs). See Supplementary Figure S1 for map showing well names. (b) Zoomed in schematic of the mapped VZDs and the eight 3D seismic reflection surveys used. (c-d) Uninterpreted and interpreted variance time-slices showing the VZDs correspond to subtle, long, linear features; time-slices shown are at 4.5 s TWT for the Chandon, Glencoe, Centaur, Colombard, Draeck, and Viper 3D surveys, but at 3.5 s TWT for the Thebe and HEX03A surveys. The nine 2D seismic reflection surveys containing observed VZDs and used to tie VZD traces between 3D surveys are also shown.**
**Yellow bars in (d) highlight section locations shown in Figures 6 and 7.**

**Figure 5: Interpreted 3D view of vertically exaggerated (VE) seismic reflection data, which images parts of VZDs D and E and highlights recorded measurements: $w$ = VZD width; $h$ = VZD spacing; $s$ = VZD segment strike; $l$ = VZD segment length (see Supplementary Fig. S2 for uninterpreted version). Note the channel on the plan-view variance time-slice is not laterally offset**
**where it is cross-cut by the VZDs. Depth shown in seconds two-way travel-time (s TWT). See Figure 4c for location. Inset top-left: plan-view sketch depicting the tip-to-tip length ($L$) and strike ($S$) measurements for an entire VZD. Inset bottom-right: schematic diagram showing how a VZD's geometry may not correspond to the true shape of the structure, or structures, it represents.**

**Figure 6: (a-f) Interpreted seismic sections from different surveys demonstrating the variations in VZD expression. The near Top**
**Mungaroo horizon (TM) and near Base Cretaceous unconformity (BC) are shown. Normal faults bounding graben, which occasionally contain pipe-like features, occur directly above and converge on VZD upper tips; these VZD-related faults are shorter and accommodate less throw relative to larger tectonic faults. For clarity, fault displacement arrows are omitted. See Figure 4d for line locations. See Supplementary Figure S4 for uninterpreted version.**

**Figure 7: (a-c) Interpreted seismic sections from different surveys demonstrating the variations in VZD expression. See Figure 4d for line locations and Figure 6 for key. See Supplementary Figure S5 for uninterpreted version.**





**Figure 8: (a) Two-way time structure map and seismic sections showing linear structures (e.g., fluvial channel boundaries?) within the Mungaroo Formation are cross-cut but not laterally or vertically offset by VZD F. The structure map is of the stratigraphic horizon interpreted in the seismic sections. (b) Variance map and seismic sections showing fluvial channels within the Mungaroo Formation are cross-cut but not laterally or vertically offset by the VZD F. The variance map is of the stratigraphic horizon interpreted in the seismic sections. See Figure 4C for locations of (a) and (b).**

**Figure 9: (a-b) Interpreted seismic sections showing the deviated trace of well Chester-1 ST1 intersecting VZD H.1. Also highlighted are the top and base of the basalt interval intersected, and its corresponding seismic horizon. See Figure 6 for key and Figure 9c for line locations. Uninterpreted sections provided in Supplementary Figure S6. (c) Two-way time structure and root-mean squared (RMS) amplitude maps for the horizon corresponding to where Chester-1 ST1 intersects the basalt interval. See Figure 4c for location. (D) Well log and lithological data from Chester-1 ST1 (Childs et al., 2013).**

**Figure 10: (a) Plot of VZD line length and rose diagram of VZD tip-to-tip strike. (b) Plot of VZD segment length and rose diagram of VZD segment strike. (c) Cumulative frequency plots of VZD line length ($L$) and segment length ($l$) to assess whether data fits normal, log-normal, negative-exponential, or power law distributions. Best-fit trendlines for both datasets reveal they conform with log-normal or negative-exponential distributions. If the curved sections of the data distribution on the power law plot are discounted, the VZD $L$ and $l$ data display a straight-line with a C exponent of 1.29 and 2.85, respectively.**

**Figure 11: (a) Plot highlighting VZD upper tip depth, measured along transects shown in the inset map, remains relatively consistent between 4.5±0.45 s TWT to 3.4±0.34 s TWT from south to north. Error bars are ±10%. (b) Plot depicting how VZD width changes from south to north. Error bars are ±50 m. Approximate (approx.) location of boundaries between the 3D seismic surveys are shown. (c) Plot depicting how VZD spacing changes from south to north. Error bars are smaller than data symbols. (d) Cumulative frequency plots showing VZD spacing is best described by a negative-exponential distribution.**

**Figure 12: (a-b) Uninterpreted and interpreted time-structure maps showing faults developed along the near Top Mungaroo Formation relative to the location of underlying VZD traces. Yellow bars in (b) correspond to seismic section locations in Figures 6 and 7; see Figure 4D for section labels. For clarity, downthrow markers are omitted. (c) Uninterpreted and interpreted 3D view of the near Top Mungaroo Formation in the Chandon 3D survey. For clarity, only VZD-related normal faults are interpreted, in addition to underlying VZD traces and sub-circular depressions (i.e. VZD-related pits).**

**Figure 13: 3D view of the top of a sub-circular depression, developed above VZD F, expressed on an Intra-Dingo Claystone horizon. The sub-circular depression is underlain by a vertical pipe-like structure, which extends down to VZD F and contains stratigraphic reflections that are offset downwards relative to their regional trend.**

**Figure 14: Schematic showing how the deviation inclination and direction of the Chester-1 ST1 borehole can be used to estimate true dyke thickness assuming the dyke walls are parallel to the VZD H.1 boundaries. By taking the intersected thickness (48 m) of the dyke and the inclination of the SE-dipping deviated well trace (18° from vertical), relative to the W-dipping (80°) VZD, we can use trigonometry to determine the distance between the dyke wall and well intersections, on a plane orthogonal to the dyke walls (i.e. 100° from vertical), is 22 m. This information, coupled with the difference between the VZD strike (093-273° and well azimuth (146°), allows us to determine the true dyke thickness is ~18 m.**

**Figure 15: (a) Near Base Cretaceous unconformity time-structure map from the western sector of the Exmouth Plateau, with interpreted dyke-induced normal fault traces (dashed lines) and pit craters (circles) highlighted (modified from Velayatham et al.,**



**2018). See Figure 15C for location. (b) Interpreted seismic section showing the cross-section structure of possible dyke-induced normal faults and pit craters (modified from Velayatham et al., 2018). See Figure 15a for location. (c) Map of dykes interpreted in this study and those perhaps marked by possible dyke-induced faults and pit craters in the western sector of the Exmouth Plateau**
**(see also Velayatham et al., 2018). The interpreted dykes broadly define a radiating swarm, across at least a 39° arc, centred on a focal area on the Carnarvon Terrace on the Cuvier Margin. We note the orientation of VZD K fits poorly with the radiating geometry of the rest of the dyke swarm, but if it is part of the Exmouth Dyke Swarm we suggest the swarm could extend across a ~54° arc.**

**Figure 16: Equal area, lower hemisphere stereographic projection of poles (yellow circles) to all measured VZD segments. Dyke pole data is contoured assuming a measured dip error of 10°; data plotted in Stereonet 10.0 and contoured using the Kamb contouring method with an interval of 1 and a significance level of 5. The minimum principal stress axis ($\sigma_3$) was defined as the centre of the dyke pole cluster, with the geometry of the cluster used to distinguish which of the two orthogonal axes were $\sigma_1$ and $\sigma_2$ (Jolly and Sanderson, 1997).**

**Figure 17: (a) Tectono-magmatic elements of the North and South Carnarvon Basins, including the inferred extent of the Exmouth Dyke Swarm and its focal area, overlain on a map of total magnetic intensity grid (EMAG2v2). Also highlighted is a proposed plume conduit site (Rohrman, 2015) and location of a large, mafic intrusion (Müller et al., 2002). Tectonic elements highlighted include: EXP = Exmouth Plateau; DSB = Dampier Sub-basin; BSB = Barrow Sub-basin; ESB = Exmouth Sub-basin; CT = Carnarvon Terrace; MSB = Merlinleigh Sub-basin; PS = Peedamullah Shelf; GP = Gascoyne Platform; BP = Bernier Platform;**
**HS = Houtman Sub-basin; WS = Wallaby Saddle; QR = Quokka Rise; CRFZ = Cape Range Fracture Zone; and the WZFZ = Wallaby-Zenith Fracture Zone. (b) Interpreted seismic section across the large mafic intrusion highlighted in Figure 17a.**

**Figure 18: Schematics depicting the magmatic evolution of the study area during the Late Jurassic. (a) Initial igneous activity led to development of a high-velocity body at the base of the crust and synchronous uplift (i.e. horizontal $\sigma_3$) and erosion to form the Callovian unconformity. (b) As emplacement of the high-velocity body waned, uplift transitioned to subsidence, marked by a rotation to a vertical $\sigma_3$ and intrusion of sill-complexes. (c-d) A renewed phase of magmatism and uplift rotated $\sigma_3$ to a horizontal orientation that favoured formation of the Exmouth Dyke Swarm. The Exmouth dyke swarm may have been fed directly from a mantle plume (c) or a large volcanic centre (d).**








**Table 1: VZD Length and Strike data**

| Name | Length (L) [km] | Strike (S) (°) |
|---|---|---|
| A | 007.1 | 021 |
| B* | 074.1 | 002 |
| C | 106.4 | 014 |
| D | 084.5 | 012 |
| E | 066.4 | 012 |
| F | 147.0 | 012 |
| G* | 170.7 | 013 |
| H* | 157.0 | 014 |
| I | 125.8 | 017 |
| J | 035.9 | 014 |
| K | 054.1 | 004 |
| L | 056.4 | 020 |
| M | 017.0 | 017 |
| N | 008.6 | 012 |
| O | 048.2 | 012 |
| P | 021.1 | 175 |
| Q | 082.7 | 178 |
| R | 019.0 | 175 |
| S | 017.9 | 002 |
| T | 027.3 | 005 |
| U | 042.5 | 017 |
| V | 003.5 | 007 |
| W | 013.1 | 001 |
| X | 013.5 | 004 |
| Y | 037.7 | 173 |
| Z | 027.6 | 001 |

*total values encompassing all physically unconnected segments of these VZDs



Figure 1

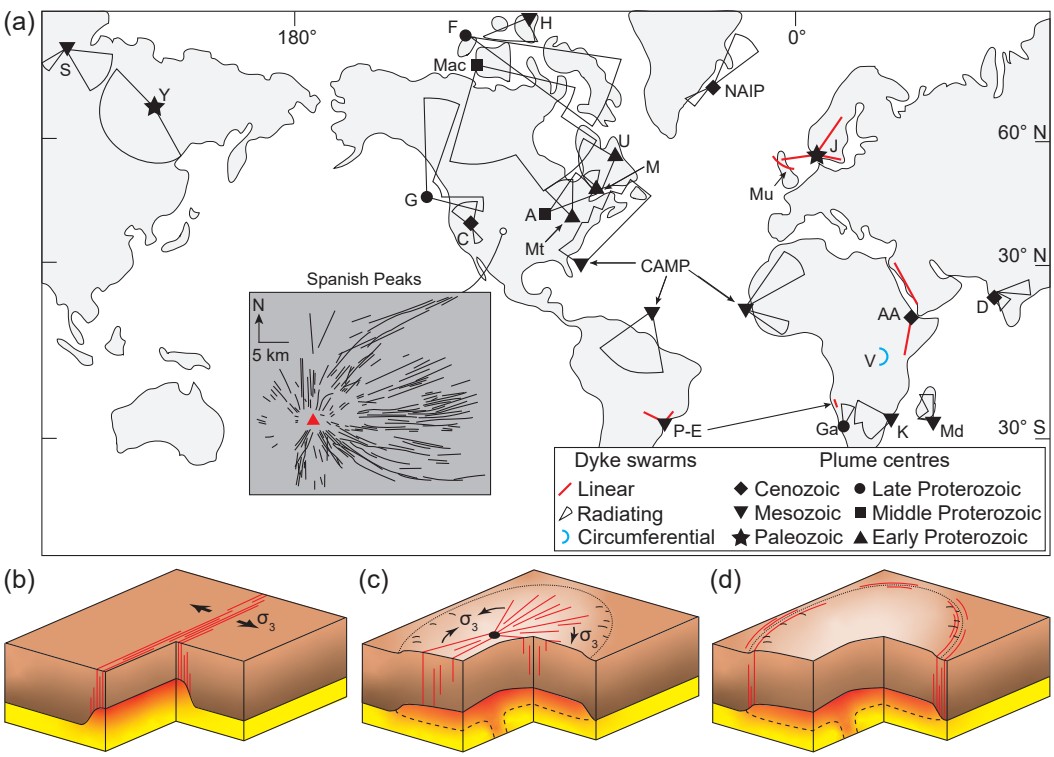



Figure 2

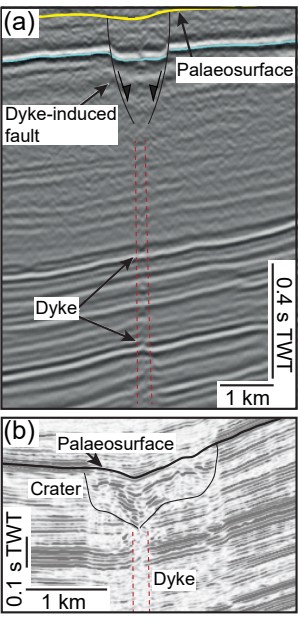





Figure 3



Figure 4

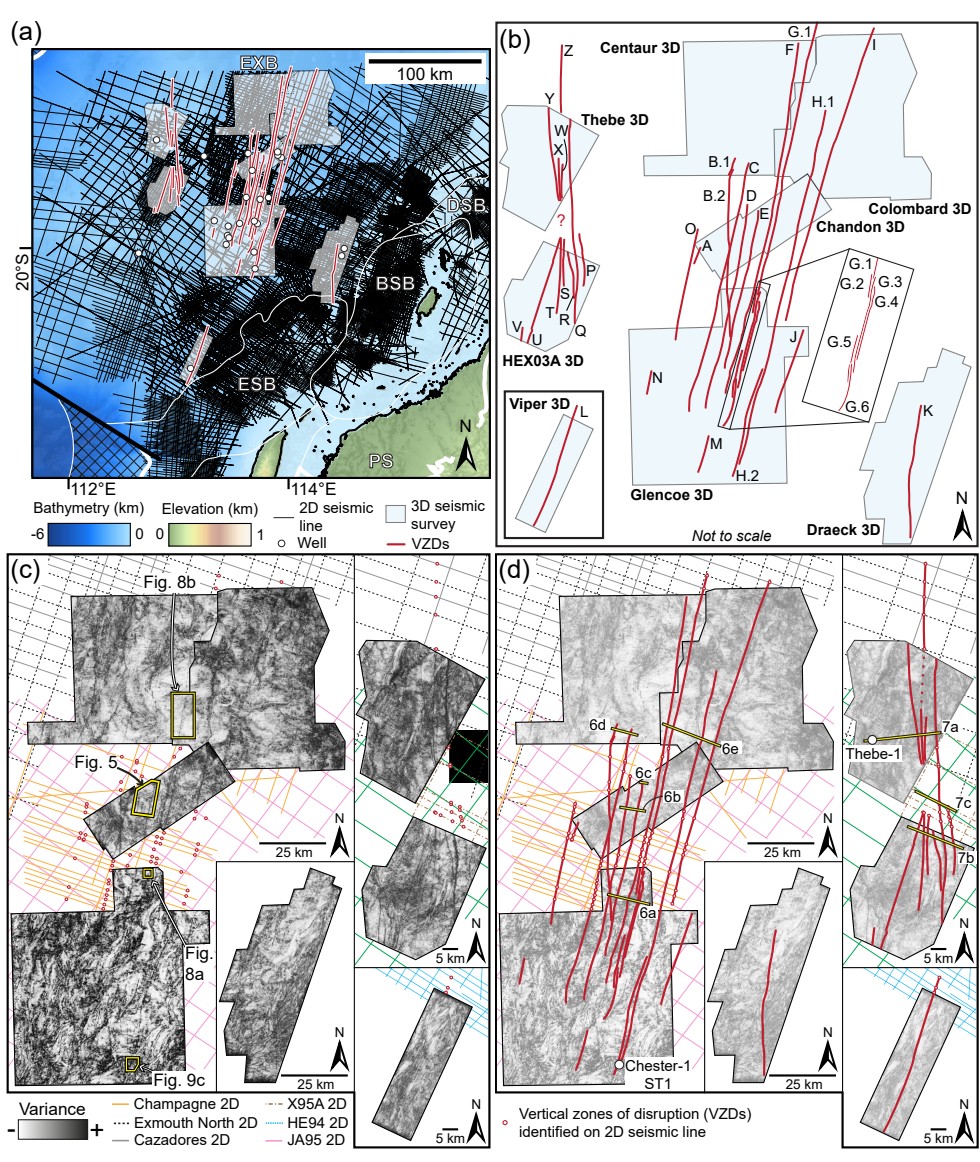





Figure 5

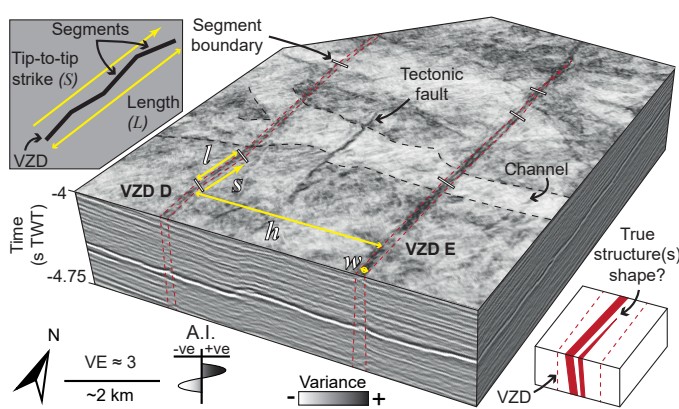


Figure 6

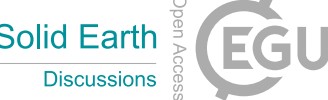

Figure 7

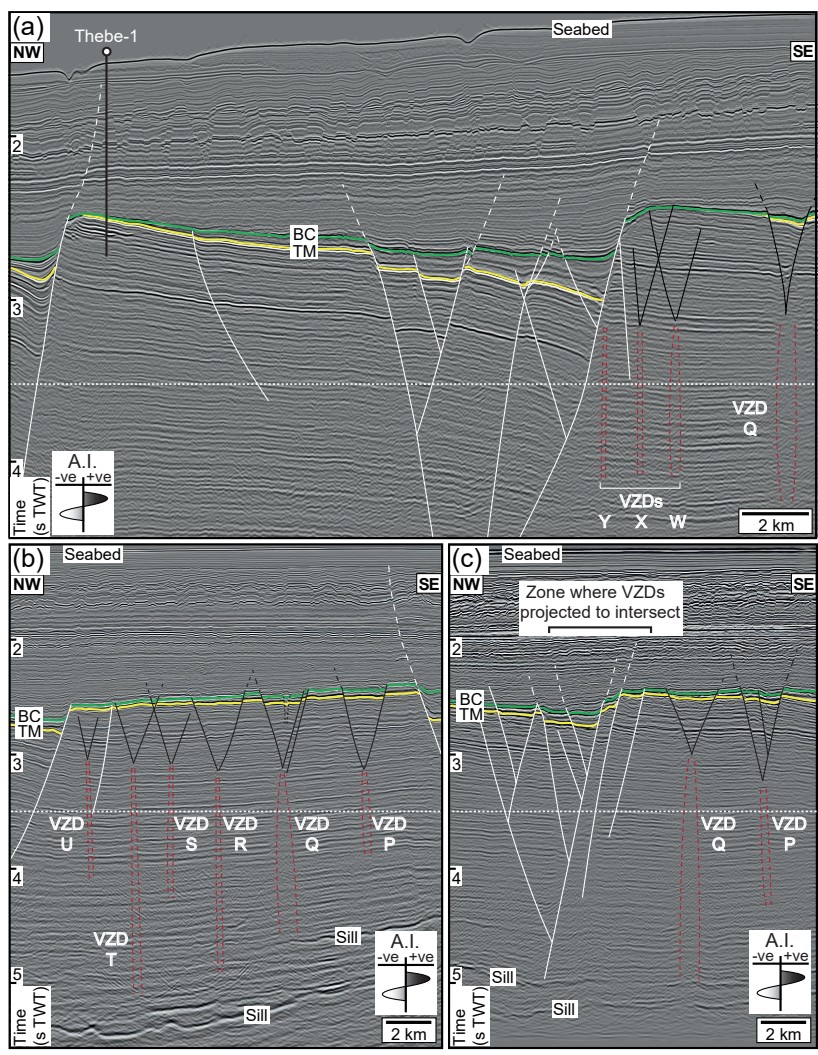


Figure 8

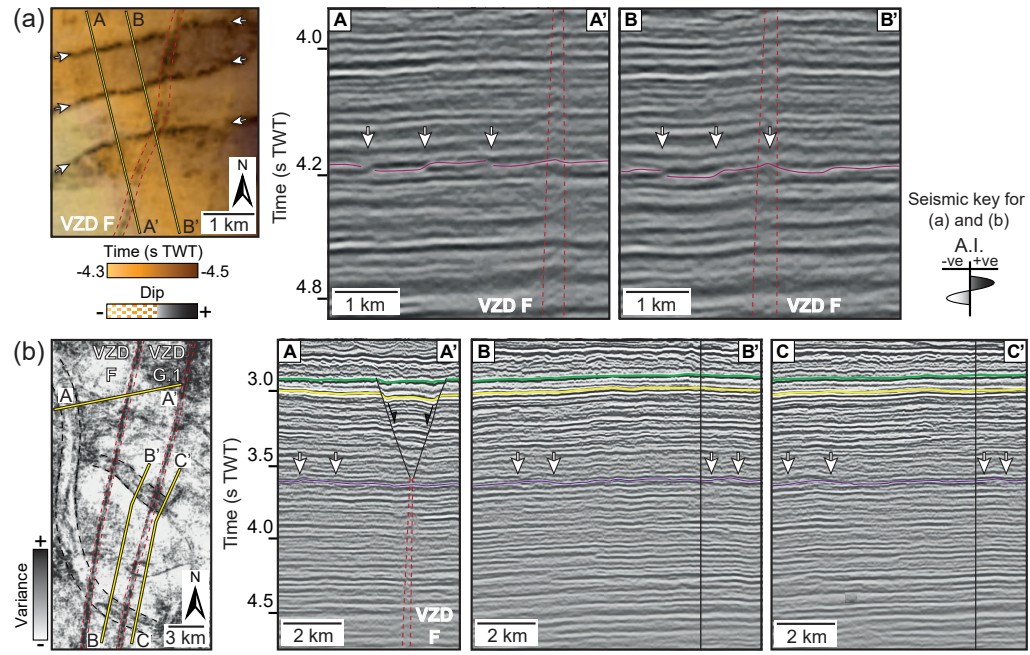

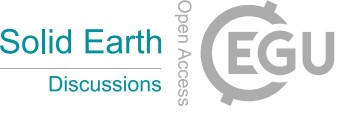
Figure 9

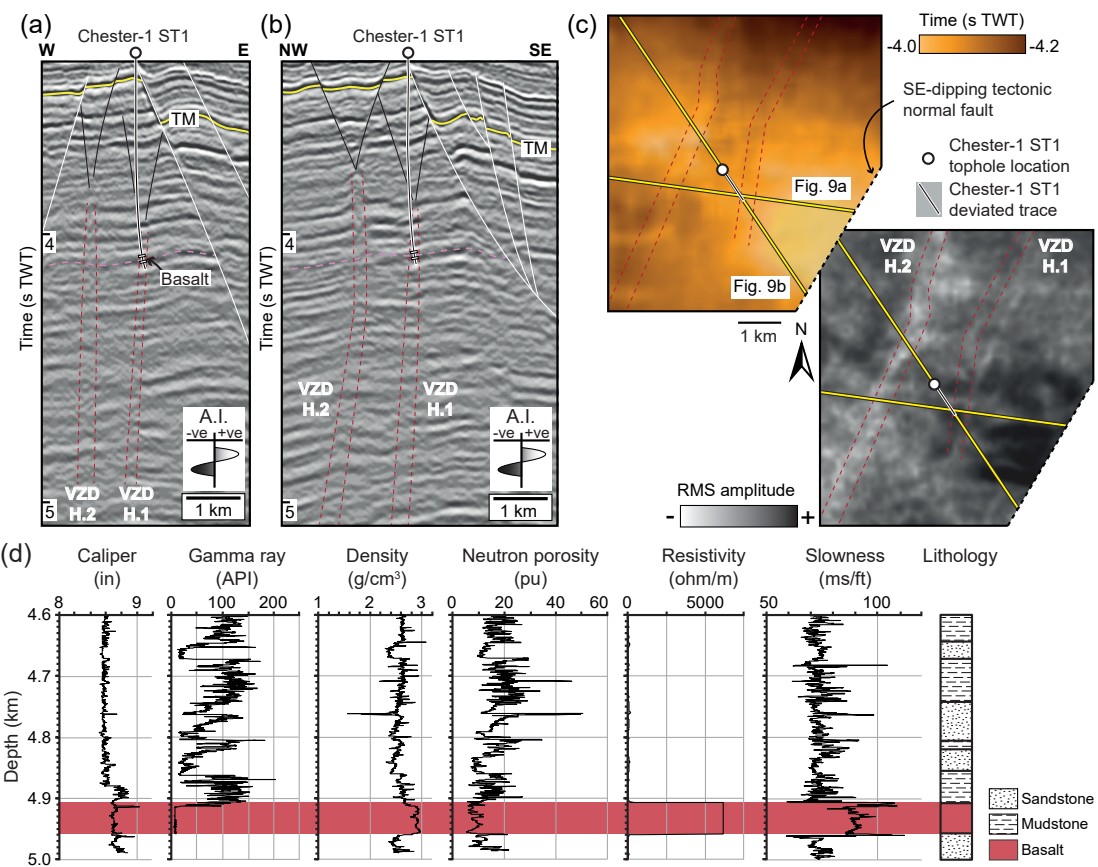



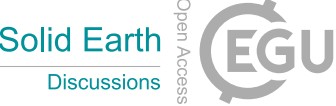

Figure 10

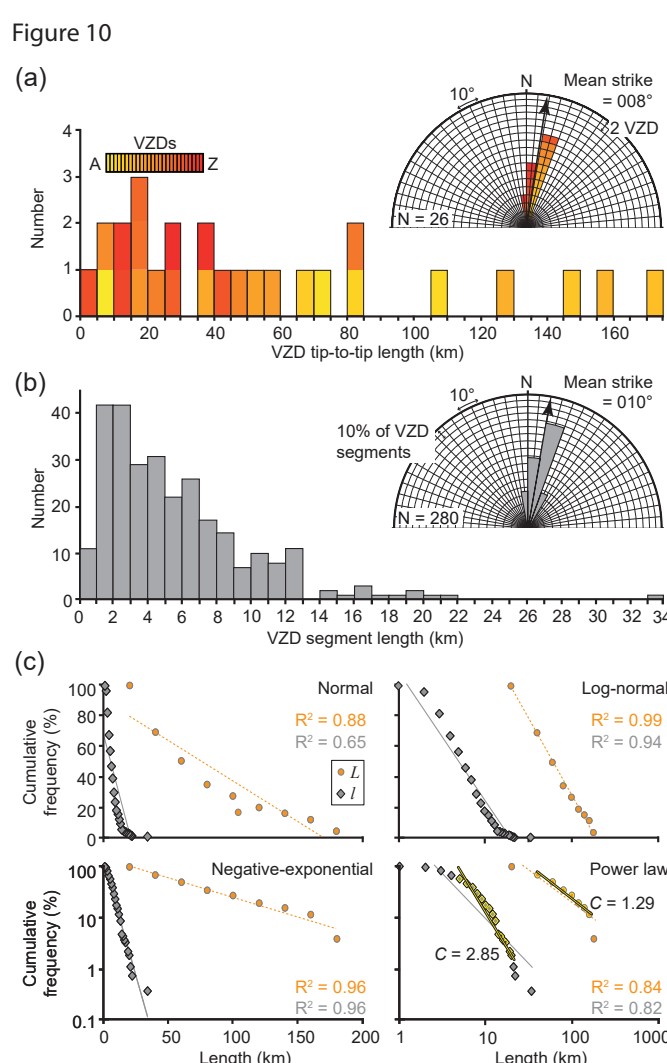





Figure 11

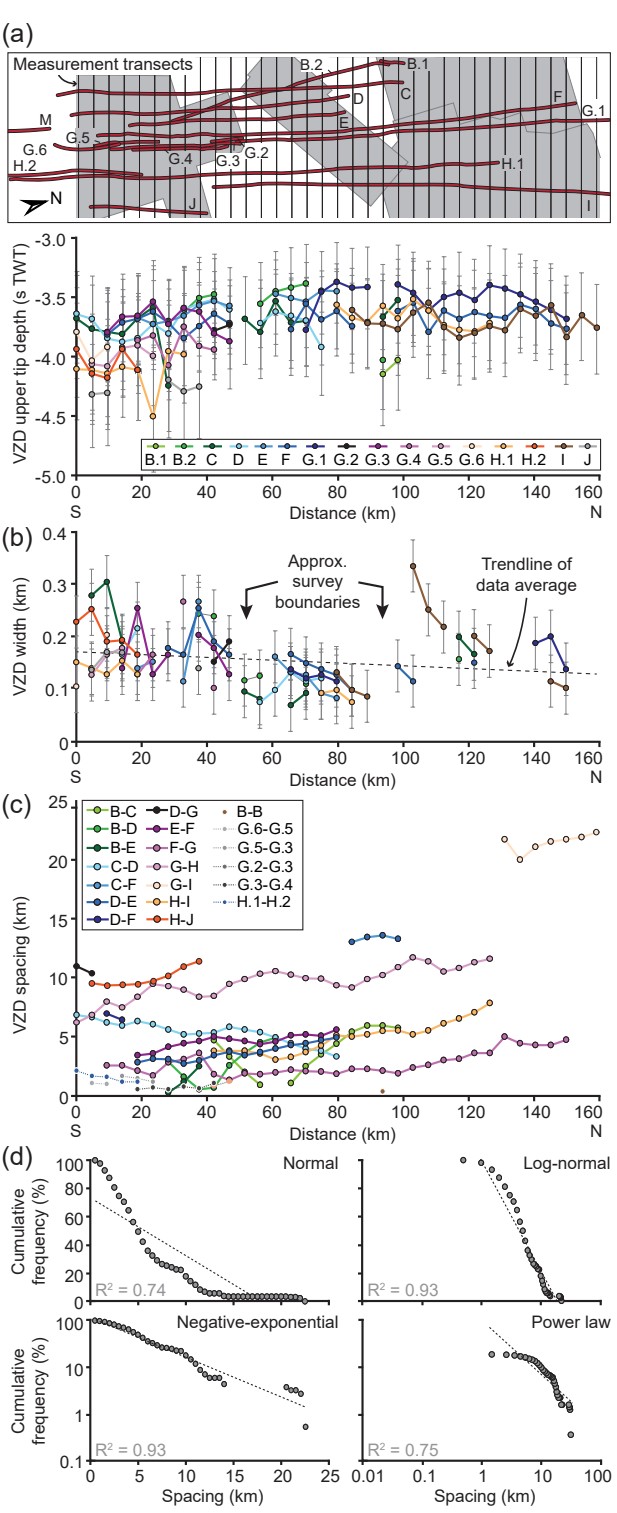




Figure 12

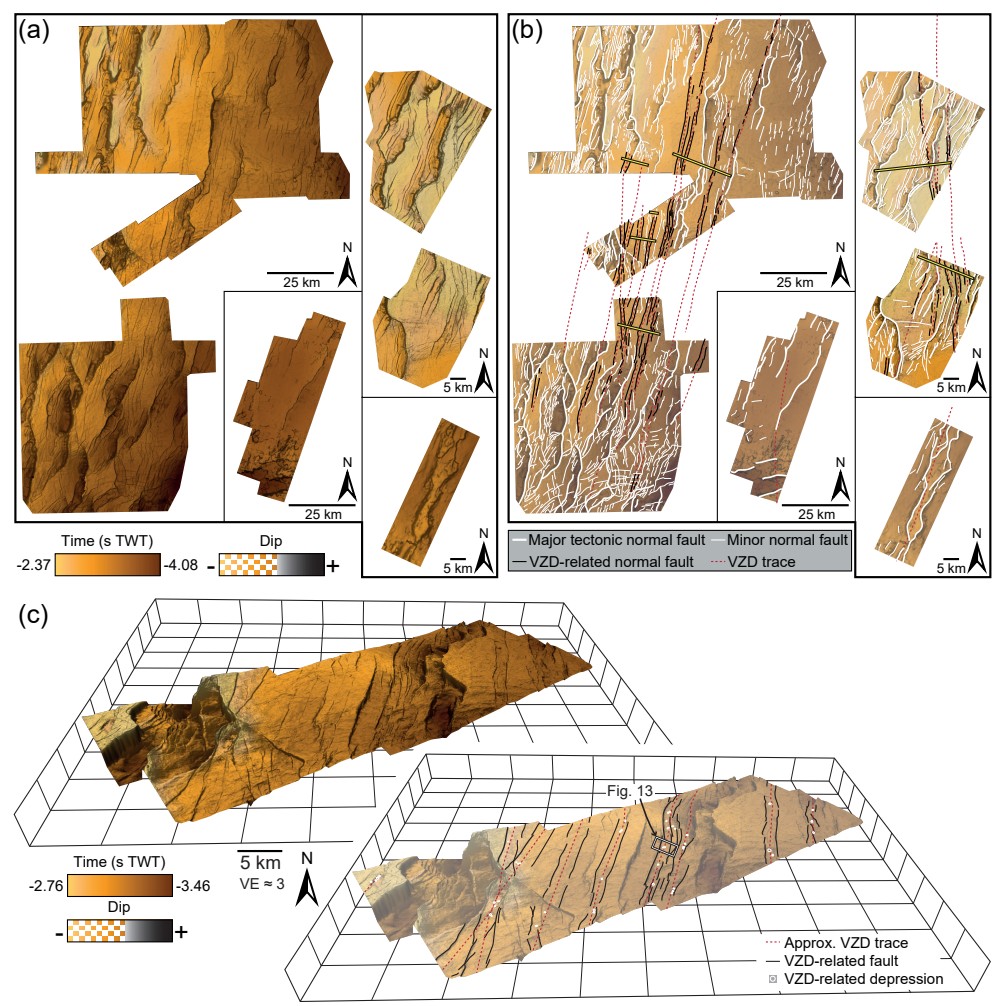



Figure 13

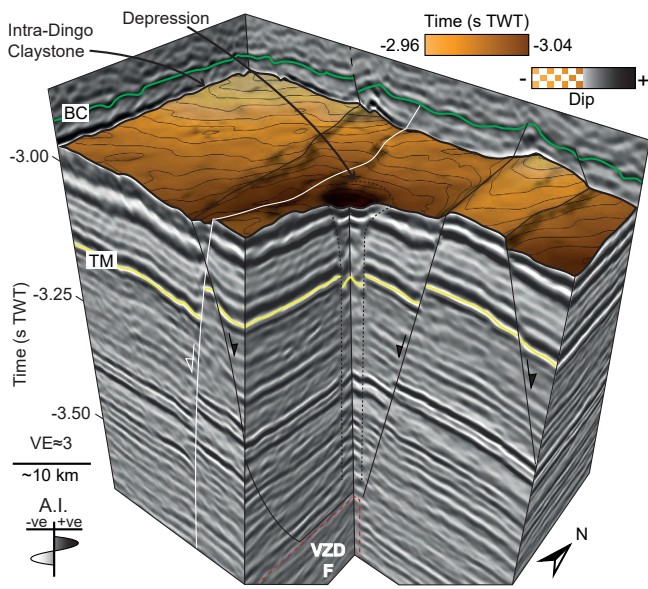



Figure 14

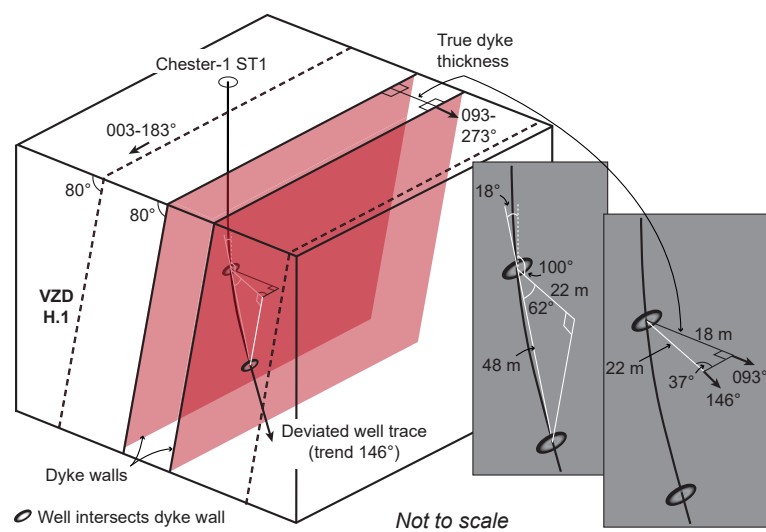



Figure 15

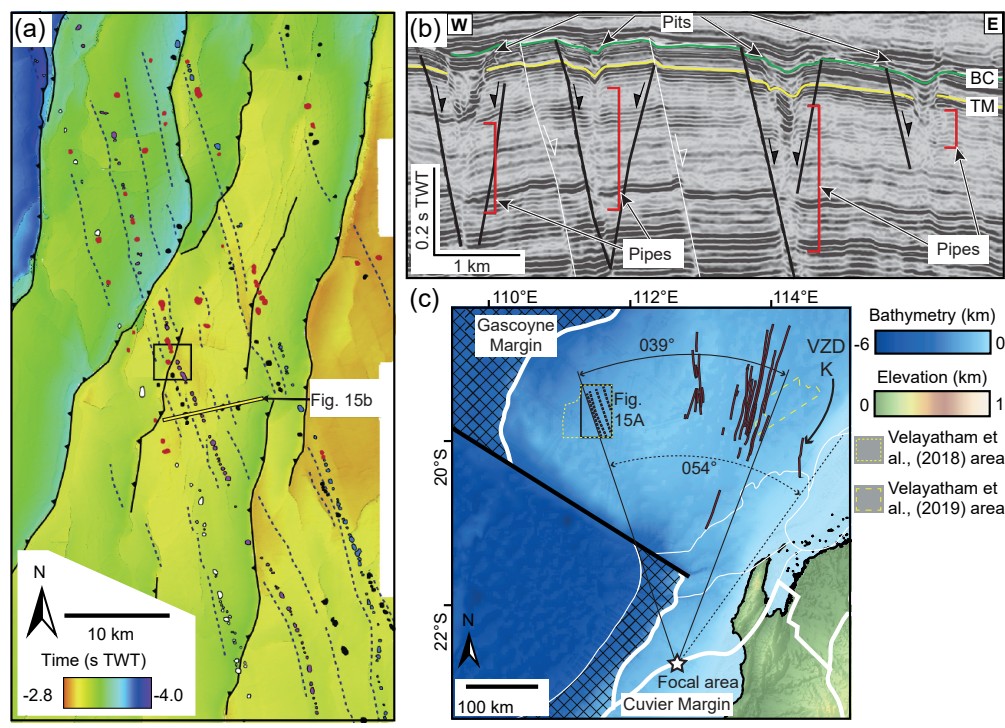

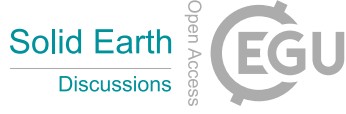

Figure 16

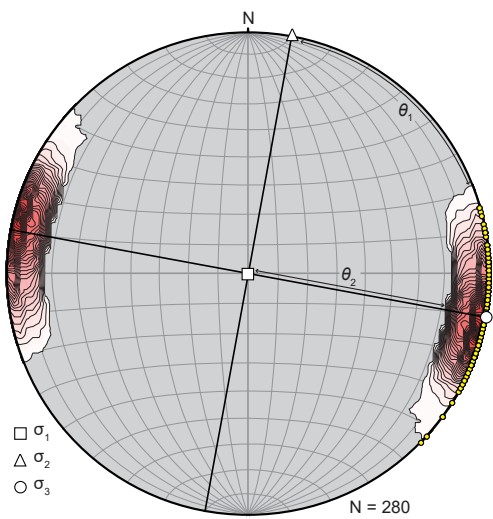



Figure 17

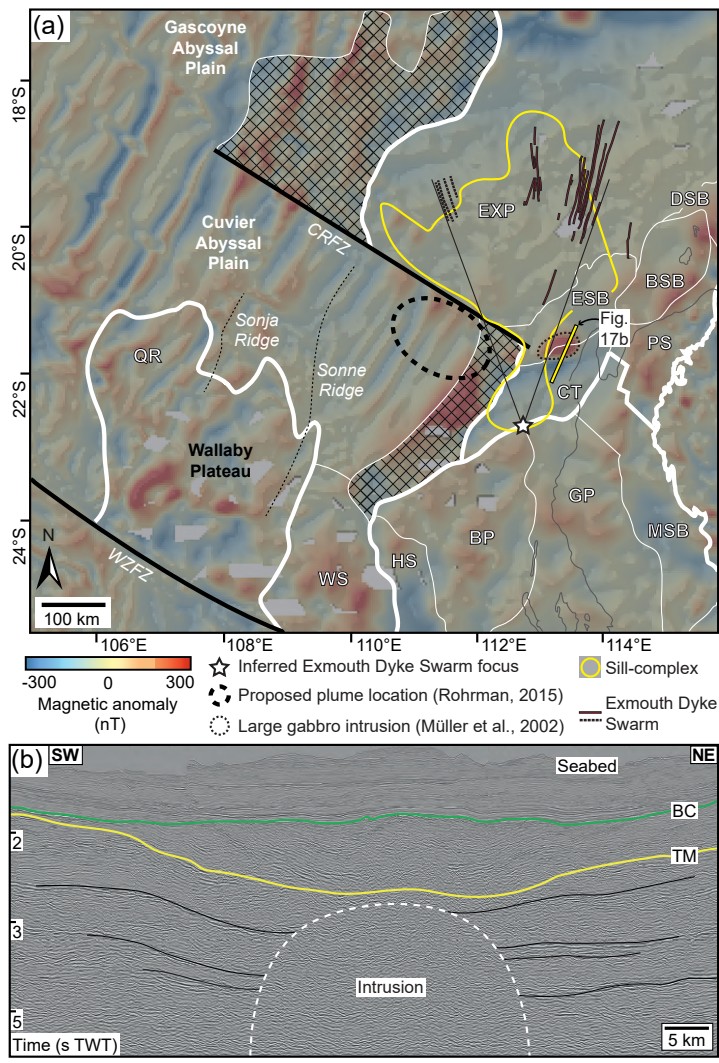



Figure 18

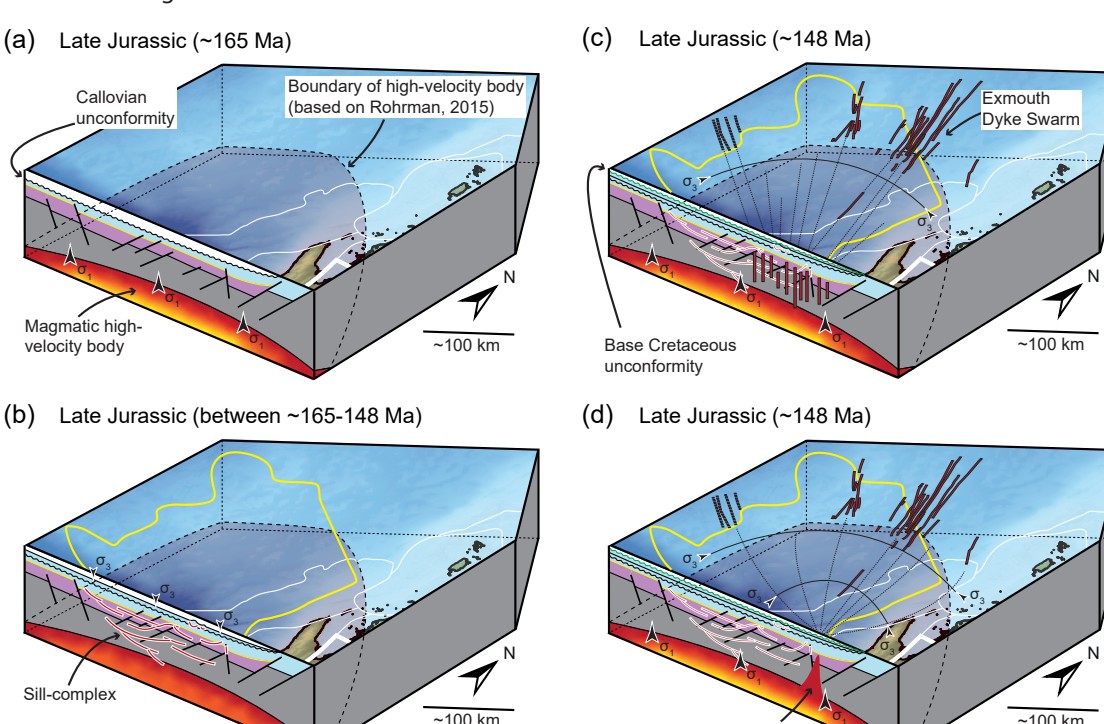