# Peer review of "Seismic reflection data reveal the 3D structure of the newly discovered Exmouth Dyke Swarm, offshore NW Australia"

_Solid Earth, 2019_

## Referee Comment (RC1) · Jonathon Hardman (Referee) · 9 Feb 2020

This paper describes, in detail, vertical zones of distribution located on the Gascoyne Margin offshore NW Australia, ultimately concluding that they are likely to be the expression of a late Jurassic dyke swarm, here named the Exmouth Dyke Swarm. The distribution of this swarm and the tectonic setting of the margin during the Jurassic are used to argue in favour of a mantle plume being the source of the dykes.

Overall, I enjoyed the paper and found it to be written to a high standard. Additionally, the paper is illustrated well and both these points are reflected in the small number of comments I have to make regarding the paper. Nonetheless, I have a few lingering

questions upon finishing the paper. Should these questions be addressed, I would have no problem recommending that the paper be accepted.

—Errors— I'm glad that errors have been taken to account throughout the paper and section 3.3 was well written, clearly outlining the errors that could contribute to the measurements within the paper. However, it was unclear to me how ±10% was assigned for errors in time and ±50 m for distances. Could the authors clarify why these numbers were chosen (presumably closely related to resolution)?

—VZD Heights— It is mentioned in section 3.1 that defining the base of the VZDs can be troublesome due to the variable data quality of the surveys. I was left wondering whether higher quality surveys have an easier to define VZD base where they can be related to structural features? Furthermore, to what extent do the authors think it's possible to constrain whether VZD height is related to imaging at depth or a geological process? Lastly, are the heights measured reasonable when constraining a source for the VZDs? In summary, it would be great if the authors could elaborate on whether their measurements of the height of the VZDs are geologically plausible and whether they are consistent throughout the study (i.e. if survey quality has an accountable effect on height estimation)

—Data Resolution and the Quantity of Dykes— Taking into account the resolution of the seismic, is the number of dykes mapped within the Gascoyne Margin comparable to the number of dykes observed in onshore swarms? A comment regarding the number of dykes not imaged might be appropriate as a caveat to the calculated dyke spacings.

—Evidence for a Mantle Plume— Have other studies attempted to map the amount of denudation prior to deposition of the Barrow Group? If erosion increases towards the south of the area that could be further independent evidence for the presence of a plume during the Jurassic.

—Specific Comments—

Line 121 – There is also some more recent work that has been conducted on the area: Mark, N.J., Holford, S.P., Schofield, N., Eide, C.H., Pugliese, S., Watson, D.A. and Muirhead, D., 2019. Structural and lithological controls on the architecture of igneous intrusions: examples from the NW Australian Shelf. Petroleum Geoscience.

Line 164 – I had to look up what Weibull distributions are and it appears they can be quite variable. Could you clarify what you mean by this statement? Is it referring to a shape or the statistics of the range of dyke thicknesses?

Line 170 and Figure 5 – I found the l, s, h and w difficult to read when overlaid on the seismic, could you make these more visible? Also, for the strike of the VZDs, could the angle you are referring to be made clear on the image? Currently, it looks quite similar to the tip-to-tip length.

Figure 1 – I found the depiction of the radiating dyke swarms to be slightly unclear, particularly in northeast America. I wonder whether shading of the radiating swarms could help make them clearer.

Figure 3 – Considering the detail exhibited in the image, I find Figure 3c to be too small (although it does highlight the key geological features for the paper. I would like the figure to be larger, particularly so that the Turonian and Intra-Hauterivian unconformities are easier to see.

—Technical Corrections—

The referencing throughout the paper was neither chronological nor alphabetical. Is it journal standard to adhere to one of these?

Line 16 – I found the choice of 'latest' to be odd as my initial thought was that it refers to multiple dyke swarms in the Jurassic. Would Late be sufficient?

Line 150 – Should "…we were able to" be the start of a new sentence?

Line 940 – Form should be from

Figure 16 is lacking a colour bar and key for the yellow points on the east of the stereonet.

---

## Referee Comment (RC2) · Janine Kavanagh (Referee) · 9 Feb 2020

General Comments

The topic of this paper is the study of a newly discovered dyke swarm offshore NW Australia. The paper uses the seismic reflection method to elucidate the lateral extent, depth, thickness and spacing of dykes identified in 26 VZD's (vertical zones of disruption). The study is significant because of the newly identified dyke swarm and the implications that the timing and extent of this magmatism have on the geological history of the area.

[Figure]

The study is exceptional in its detail, the figures are excellent and the text is very well written. I have some suggestions on how the manuscript could be improved, particularly with respect to quantifying strain, magma volumes, describing errors and discussing how far the measurements can be taken to inform interpretations. I also make a few suggestions about additional context and assumptions behind statements which I think should be explained more fully.

Overall, I find this to be an interesting paper and case study with an exceptional dataset. If my comments can be addressed then I would be happy to support its publication.

Best wishes,

Janine Kavanagh

Specific Comments

A) Geometric dimensions and magma volumes:
1) The word 'length' is used but it took me a while to understand clearly what direction that was exactly. 'Width' is also used and then 'thickness' sometimes too. Please use these words consistently throughout and give a clear definition at the start. I suggest adding 'vertical' or 'horizontal' to the word 'length' to make it without doubt which direction you are describing. Also I suggest always using 'thickness' for the shortest dimension is described. This is then more consistent with existing geological publications of dyke datasets too.
2) Is it possible to give an estimate of the dyke-related magma volume in the area?

How significant is this in the geological history?

B) Strain estimation:
1) How much strain has the dyking accounted for across the area? How does this compare strain accumulated due to spreading rates during the active period (based on your 'timings' constraints)? Comparing these rates would perhaps enable you to comment on if the dykes were overpressured or not.

C) Impact of errors and limitations on interpretations:
1) The error quantification really seems key to what can and cant be said in this study. It apears the VDZ thickness (for example) generally is thicker than dyke thickness, so please state this early on in the paper. What is the composition of these dykes? Can you use the general thicknesses of dykes in sedimentary basins to see how the VDZ thickness compares? Seems strange to place errors as '+/-' without a bit more explanation.
2) It seems likely VDZ thickness overestimates dyke thickness quite substantially? Is it likely to be consistent e.g. as

D) Applications to other sites:
1) I think something that is missing from the text is a comment about what your study means for reinterpreting datasets where dykes may have intruded and yet cant be imaged? Is there an opportunity to state what proportion of magmatism might be underestimated in relevant comparative regions?

Technical Corrections

Abstract
Line 8 – 'extend laterally for..'

Line 9 – 'their presumed rapid emplacement,..'
Line 16 – can you give an indication of the quantification of the detail? To what quantity of resolution?
Line 16 – what do you mean the latest? Relative to what?
Line 17 – the word 'long' needs some context. Is it the horizontal 'length'? The vertical 'length'? Length generally implies to the longest dimension. Are these vertical dykes or blade-like (horizontal) dykes? How deep do they extend?
Please give a general overview of the measurements made.

Introduction
Line 25 – please define 'rapidly' quantitatively (or at least give a range of timescales).
Line 27 – 'We recognise' – I think you mean in the literature there are 3 dyke swarm geometries. Please rephrase to take yourselves out of the writing.
Line 34 – '..emplacement is thought to be primarily..'
Line 34 – 'extending the host rock rather than through magmatic overpressure'.
Line 36 – 'drive crustal extension, influencing...' – I wonder if this really can be stated based on the observations, or could it be said these dykes are a consequence of crustal extension? Seems the latter if the magma passively fills fractures.
Line 39 – 'syn-emplacement stress conditions' – please clarify that these stresses could be local or regional
Line 48 –magnetic surveys provide insight too.
Line 53-54 – Kavanagh and Sparks (2011) data also describes vertical variation in dyke geometry, as well as the lateral variation you attribute.
Line 54-57, 67-68 - Phillips et al. (2018), Magee et al. (2019) – how is your paper different to these? Please clarify.
Line 72 – 'length' – this needs some orientation in space. Lateral? Vertical? Not both?

Section 2
Please state the composition or compositional range of the magma.

Section 3

Line 161 – "controls and is reflected in" – isn't this the same statement repeated? Either 'controls' or 'reflected in' would be sufficient. I don't see the difference.

Line 162-163 – "dyke lengths" – I am confused by what you mean by 'length'. Vertical or horizontal dimension? Clarifying this at the start of the paper would be helpful.

Line 164-165 – "follows a Weibull distribution" – can this point be made as generally as this? These are just two studies you point to, and actually the suggestion is primarily made in the Krumbholtz paper only. Please give some context to the statement. I don't think the statement can be generally made based on just these two papers cited.

Line 166 – 'dyke geometry and distribution' seems more appropriate than 'dyke properties'

Line 170 –remove comma 'plan view tip-to-top length (L)'

Line 170 –now I understand what you mean by 'length'. I think it would be clearer if you rephrased throughout the paper as 'horizontal length' to be clearer.

Line 173 - Ditto with regard to the use of 'width'. In the other datasets you referred to in the literature review this dimension was called 'thickness' so please be consistent with this and rephrase.

Line 198 – 'VZD thickness' would be clearer (rather than relating a width to a thickness, relate a thickness to a thickness)

Section 5

Line 323 – "We therefore consider it unlikely that the VZDs are faults." If these dykes have not intruded existing faults, then does this suggest the dykes created there own fracture during propagation and were overpressured?

Section 6

Line 391 – 'breadth, thickness and spacing'

Line 396 – 'Seismic reflection data thus provide a unique opportunity to examine and

quantify the 3D structure of a dyke swarm independent of the potential bias introduced by the processes (e.g., erosion) controlling how dyke swarms intersect the surface.' I feel this is overstated. It is possible to access different depths of dyke swarms in the geological record by studying areas at different palaeodepths. The thickness increase of dykes with depth based on seismic data seems too speculative at the moment given the errors associated with the method (particularly in that dimension).

510 – I am not yet convinced that subtle changes in VDZ thickness can be related to dyke thickness changes. The errors appear to be too large.

513 – there is ongoing discussion about the Bardabunga dyke swarm and the origin of the magma. The lateral propagation is a hypothesis, however there is also evidence suggesting vertical propagation. So I don't think Bardabunga can be used so decisively to state dykes in your area propagated laterally.

Figures
Generally of the highest quality. Excellent.
Figure 1 - I like this figure but please remove the 'magma pond' at the base of b-d as it is not needed in your figure and is speculation on the nature of the magma source (suggesting it were entirely molten at once and was a large reservoir, whereas it may also have been transient small batches of distributed melt).

---

## Author Comment (AC1) · 19 Mar 2020

Dear Editor,

We are extremely grateful to Jonathon Hardman and Janine Kavanagh for the positive, constructive reviews. We consider their comments have helped improve the manuscript and we address each point below:

Reviewer 1: Jonathon Hardman

1) Errors - I'm glad that errors have been taken to account throughout the paper and section 3.3 was well written, clearly outlining the errors that could contribute to the

measurements within the paper. However, it was unclear to me how ±10% was assigned for errors in time and ±50 m for distances. Could the authors clarify why these numbers were chosen (presumably closely related to resolution)?

Upon reflection, we realise that most error sources (data quality, resolution, and depth conversion) discussed in Section 3.3 do not actually impact the measurement of VZD thickness, length, or depth. Rather, they control VZD seismic expression: i.e. there are two things to consider, (1) the imprecision of measuring VZD geometry, and (2) the errors associated with relating our VZD measurements to dyke properties (this latter point is the focus of the Discussion section). We thus consider that any errors in our VZD measurements are only introduced by human uncertainty and the precision with which we measure things. Based on personal experience, we have therefore defined these as 0.05 s TWT (measurements in depth: e.g., upper tip depth) and 50 m (measurements in plan-view: e.g., length and thickness). Our approximate depth-conversions of measurements in time are, however, subject to error in the velocities used, which we estimate to be [plus/minus]10% ; this is an arbitrary value commonly used in seismic reflection literature (e.g., Magee et al., 2013). We have modified all measurements accordingly (including on plots; i.e. Fig. 11) and clarified this in the text:

"Overall, the described data quality, resolution, and depth conversion error sources do not compromise the precision of VZD thickness, length, and height measurements. Rather, uncertainties and/or variation in these error sources are introduced when attempting to relate VZD geometry to that of the geological features they represent, which we consider in the Discussion. However, to account for potential errors introduced by human imprecision during measurement, we conservatively consider that each quantitative parameter could have an arbitrary error of either: (i) ±0.05 s TWT if the property analysed is measured in time (e.g., VZD upper tip depth); or (ii) ±50 m if distances (e.g., VZD length, width, and spacing) are measured in plan-view. These values are based on personal experience. To help geoscientists more used to work with geological (e.g. field) rather than geophysical data, and to provide an overall sense of scale, we use velocity data to provide approximate depth-converted value (in metres) for measurements in time. Due to uncertainty in the velocities used for these depth-conversions we cannot ascertain their accuracy and thus present them with arbitrary errors of $\pm 10\%$." (Lines 259-269)

We have also added the following text to the Discussion to further reinforce our decisions regarding likely errors in our analysis:

"Whilst seismic reflection data can provide unprecedented insights into the 3D structure of dyke swarms, limitations and uncertainties in seismic and/or borehole data quality, resolution, and depth-conversion make it difficult to relate the quantifiable VZD seismic expression to the true geometry of the dykes they likely represent. For example, we cannot resolve whether a mapped VZD, even if it is intersected by a borehole (e.g., Fig. 9), corresponds to a single, or multiple closely spaced intrusions. Here, we specifically discuss how our VZD measurements can be used to evaluate how dyke length, thickness, and spacing may compare to predicted distributions of these geometrical properties based on surface- and physical-, numerical-, and analytical modelling-based studies." (Lines 529-535)

2) VZD Heights - It is mentioned in section 3.1 that defining the base of the VZDs can be troublesome due to the variable data quality of the surveys. I was left wondering whether higher quality surveys have an easier to define VZD base where they can be related to structural features? VZD lower tips are very difficult to define in any of the seismic surveys used. It is correct that the expression of VZDs in higher-quality surveys is better at greater depth, but towards the lower limits of any survey they become difficult to confidently define. We consider this comment is addressed by:

"Only on a few seismic sections, where data quality is high, do we observe undisturbed reflections directly beneath a VZD, thereby allowing us to constrain its height (e.g., Fig. 6C). For example, the depth to the base of VZD E appears to decrease northwards

from ≳5.8±0.05 s TWT to ∼4.4±0.05 s TWT (∼8.5±0.85–5.6±0.56 km) (e.g., Figs 6A-C)." (Lines 370-373)

3) Furthermore, to what extent do the authors think it's possible to constrain whether VZD height is related to imaging at depth or a geological process?

We would consider that the likely defined VZD bases are constrained by imaging, and as such probably do not correspond to true VZD lower tip depths. However, without better seismic data or deep borehole data, we cannot test this hypothesis. A key point here is that, in the Discussions, we acknowledge that dyke height could be up to ∼24 km (equivalent to the crustal thickness), but we highlight that our height measurements provide a minimum estimate; constraining the range of possible dyke heights allows us to more accurately evaluate spacing relationships. See response to comment below.

4) Lastly, are the heights measured reasonable when constraining a source for the VZDs? In summary, it would be great if the authors could elaborate on whether their measurements of the height of the VZDs are geologically plausible and whether they are consistent throughout the study (i.e. if survey quality has an accountable effect on height estimation)

The VZD heights we measure represent a minimum estimate for dyke height. We do not think the reviewer questions this, but rather whether the maximum estimate for dyke height (∼24 km), which is based on the crustal thickness and that we use in the dyke spacing analysis, is geologically plausible. We have added a brief discussion regarding the plausibility of these end-members and their implications:

"Because the dykes are typically >1.5±0.05 s TWT tall (e.g., Figs 6 and 7), we use extrapolated checkshot data to estimate the average H is at least ∼3.5±0.35 km (Supplementary Fig. S3). Compared to our current understanding of the different theories of dyke emplacement (Townsend et al., 2017), our minimum height estimate implies the dykes are encased within sedimentary strata and were emplaced either as: (i) ascending dykes of a fixed fluid volume, where upwards migration was balanced by

closure at its lower tip (School 1); or (ii) lateral propagation of a dyke with a fixed height (School 3). In contrast, as a maximum estimate for average H, we consider the dykes could extend upwards from a source (e.g., the high-velocity body; Rohrman, 2013) towards the base of the crust (e.g., School 2; Townsend et al., 2017), which across the Exmouth Plateau is likely ∼20–28 km beneath the present seabed (e.g., Mutter and Larson, 1989; Stagg and Colwell, 1994; Tindale et al., 1998; Stagg et al., 2004; Reeve et al., 2016). Given the upper dyke tips broadly occur at ∼3.7±0.05 s TWT, equivalent to a depth of ∼4.1±0.41 km, we therefore suggest the maximum average H could be up to ∼24 km." (Lines 660-670)

Due to variations in data quality, we cannot currently confidently assess potential variations in dyke height across the study area.

5) Data Resolution and the Quantity of Dykes - Taking into account the resolution of the seismic, is the number of dykes mapped within the Gascoyne Margin comparable to the number of dykes observed in onshore swarms? A comment regarding the number of dykes not imaged might be appropriate as a caveat to the calculated dyke spacings.

This is an important point and we have added the following:

"However, dykes swarms exposed onshore typically contain significantly more dykes than the 26 we identify in our seismic reflection data (cf. Gudmundsson, 1983; Jolly and Sanderson, 1995; Mège and Korme, 2004)." (Line 672)

6) Evidence for a Mantle Plume - Have other studies attempted to map the amount of denudation prior to deposition of the Barrow Group? If erosion increases towards the south of the area that could be further independent evidence for the presence of a plume during the Jurassic.

Such a denudation pattern as described by the reviewer here was quantified by Rohrman (2015) and used to support a mantle plume hypothesis. We have modified the following to clarify this:

[Figure]

"Any process invoked to explain the origin of a thermal anomaly in the mantle in the Late Jurassic, and potentially the Early Cretaceous, needs to account for: (i) the latest Jurassic distribution of magmatism across the Gascoyne and Cuvier margins (e.g., Mutter et al., 1988; Hopper et al., 1992; Mihut and Müller, 1998; Müller et al., 2002; Rohrman, 2013); and (ii) recognition of domal denudation patterns and formation of contemporaneous regional unconformities (e.g., the near Base Cretaceous unconformity) (Underhill and Partington, 1993; Rohrman, 2015)." (Line 775)

7) Line 121 – There is also some more recent work that has been conducted on the area: Mark, N.J., Holford, S.P., Schofield, N., Eide, C.H., Pugliese, S., Watson, D.A. and Muirhead, D., 2019. Structural and lithological controls on the architecture of igneous intrusions: examples from the NW Australian Shelf. Petroleum Geoscience.

We consider the use of 'e.g.,' highlights that there are other works that could be cited.

8) Line 164 – I had to look up what Weibull distributions are and it appears they can be quite variable. Could you clarify what you mean by this statement? Is it referring to a shape or the statistics of the range of dyke thicknesses?

Following a comment from the other reviewer, we have removed mention of dyke thickness following a Weibull distribution. First, it was not necessary to mention this in the text here (i.e. the methodology). Second, it is sufficient to say thickness and spacing distributions can provide insights into the controls on dyke emplacement.

9) Line 170 and Figure 5 – I found the l, s, h and w difficult to read when overlaid on the seismic, could you make these more visible? Also, for the strike of the VZDs, could the angle you are referring to be made clear on the image? Currently, it looks quite similar to the tip-to-tip length.

We have increased the visibility of l, s, h, and w in Figure 5, and clarified the tip-to-tip strike measurement method. See Figure 5.

10) Figure 1 – I found the depiction of the radiating dyke swarms to be slightly unclear,

particularly in northeast America. I wonder whether shading of the radiating swarms could help make them clearer.

We have shaded the radiating dyke swarms to make them more visible.

11) Figure 3 – Considering the detail exhibited in the image, I find Figure 3c to be too small (although it does highlight the key geological features for the paper. I would like the figure to be larger, particularly so that the Turonian and Intra-Hauterivian unconformities are easier to see.

To fit a ∼400 km long seismic long onto an A4 page in portrait involves reducing its clarity. Due to figure size restrictions, we cannot change do this in Figure 3c without compromising clarity. However, we have added an enlarged version of Fig. 3c in the supplementary files as an A3 landscape image (see Supplementary Figure 1).

12) The referencing throughout the paper was neither chronological nor alphabetical. Is it journal standard to adhere to one of these?

We used the Endnote output style provided by Solid Earth but note that there is no specific setting for in-text citation order. However, we have changed this so references are displayed chronologically and alphabetically for simplicity.

13) Line 16 – I found the choice of 'latest' to be odd as my initial thought was that it refers to multiple dyke swarms in the Jurassic. Would Late be sufficient?

We originally used 'latest' because we did not want to imply, by saying Late Jurassic dyke swarm, that intrusion was prolonged throughout the entire Late Jurassic. However, in hindsight and thanks to the reviewer's comments, we see how the use of 'latest' can also introduce confusion. At the reviewer's suggestion we adopt the use of 'Late' throughout the manuscript.

14) Line 150 – Should ". . .we were able to" be the start of a new sentence?

Corrected.

15) Line 940 – Form should be from

Corrected.

16) Figure 16 is lacking a colour bar and key for the yellow points on the east of the stereonet.

We have added a legend to Figure 16.
* * *
Reviewer 2: Janine Kavanagh

17) The word 'length' is used but it took me a while to understand clearly what direction that was exactly. 'Width' is also used and then 'thickness' sometimes too. Please use these words consistently throughout and give a clear definition at the start. I suggest adding 'vertical' or 'horizontal' to the word 'length' to make it without doubt which direction you are describing. Also I suggest always using 'thickness' for the shortest dimension is described. This is then more consistent with existing geological publications of dyke datasets too.

We have adopted these suggestions: width is always referred to as thickness and, where necessary, we define length as the horizontal length.

18) Is it possible to give an estimate of the dyke-related magma volume in the area? How significant is this in the geological history?

We agree that this would be a useful addition, although we highlight that the estimates are rather speculative (see also the new Table 2):

"6.2.4 Dyke swarm volume Although it is difficult to accurately constrain dyke thicknesses and heights using our data, here we use the measured horizontal length (L) of each dyke, an assumed average dyke thickness of ~20 m, and dyke heights of ~3.5–24 km to estimate dyke volumes (Table 2). If the dykes have are relatively short (in terms of their height; i.e. ~3.5 km), we estimate dyke volumes range from ~0.5–11.9

km3, whereas if the dykes are relatively tall and extend down to the base of the crust, their volumes may range from ∼3.4–81.9 km3 (Table 2). We calculate that the cumulative volume of the mapped dykes ranges from ∼102.6–703.2 km3 (Table 2). These are undoubtedly minimum values, given the likely presence of sub-seismic dykes." (Lines 688-702)

We cannot find any information on total volumes of igneous material within the broader study area, thus we cannot comment on the relative local significance of the newly discovered dyke swarm. As stated in the text (Lines 688-702; see also above), we are fully aware that our calculated volumes are only crude under-estimates, making it difficult to compare them with magma volumes of other dyke swarms. Further work is required in this area.

19) How much strain has the dyking accounted for across the area? How does this compare strain accumulated due to spreading rates during the active period (based on your 'timings' constraints)? Comparing these rates would perhaps enable you to comment on if the dykes were overpressured or not.

Strain estimation requires information concerning the true thickness of dykes. Following reviewer comments below, we have added a discussion about what the true dyke thicknesses may be and, based on several assumptions, have estimated cumulative dyke thickness and thereby associated extension:

"We show individual VZD thicknesses measured across multiple 3D seismic surveys range from 335±50 m to 68±50 m and gradually decrease northwards (Fig. 11C). Furthermore, although there are gaps in our thickness measurements where VZD imaging is locally inhibited, we estimate that cumulative VZD thickness across our selected transects also decreases northwards, from ∼1.2–0 km (Fig. 11B). Because the northwards decrease in VZD thickness is consistent across multiple seismic surveys, which each have different acquisition and processing parameters, we suggest this trend could mark a similar northwards decrease in true dyke thickness (Fig. 11B). However, synthetic

seismic forward models suggest the thickness of VZDs corresponding to sub-vertical dykes is greater than the true dyke thickness (Eide et al., 2018). Furthermore, because VZD thickness is partly controlled by the acquisition and processing properties of the seismic reflection data in which they are imaged in (e.g., frequency; Eide et al., 2018), evidenced by the marked differences in VZD thickness between different seismic surveys (Fig. 11B), it is difficult to determine how VZD thickness and true dyke thickness are related. Using observations from the Chester-1 ST1 well, which likely intersects a 48 m long section of a basalt dyke, we calculate the dyke has a true thickness of ∼18 m, assuming its orientation is parallel to that of the ∼130±50 m wide VZD it relates to (Fig. 14). These well data confirm synthetic seismic forward model predictions that dyke-related VZD thickness is, in at least some cases, much greater than true dyke thickness (Eide et al., 2018). Based on the dyke thickness constrained by Chester-1 ST1 and its corresponding VZD expression, if we consider all VZDs have thickness ratio to true dyke thickness of at least ∼7:1, we estimate dyke thicknesses measured across our selected transects range from ∼47±6 m to ∼10±6 m; we note that we cannot distinguish whether the VZDs correspond to single dykes or multiple dykes. These dyke thickness values are closer to, although typically still larger than, dyke thickness distributions measured in onshore examples where most dykes are 0–10 m thick, potentially up to 20–40 m thick (e.g., Gudmundsson, 1983; Jolly and Sanderson, 1995; Mège and Korme, 2004; Klausen, 2006; Kavanagh and Sparks 2011; Krumbholz et al., 2014). Because dykes are commonly accommodated by host rock dilation, their thicknesses are a proxy for the amount of syn-emplacement extension of an area (e.g., Jolly and Sanderson, 1995; Marinoni, 2001). We estimate the cumulative dyke thickness measured across our selected transects decreases northwards from ∼170–0 m, which given each transect is ∼51 km long and assuming dyke opening was purely dilational, corresponds to ∼0.33–0% extension; this is a minimum estimate of strain accumulation given there are undoubtedly numerous sub-seismic present in our study area. It is unknown whether this estimated extension of up to 0.33% accommodated by dyking is applicable to the entire dyke swarm. Further work in understanding how
dykes are expressed in seismic reflection data is required before these data can be used to accurately quantify dyke thickness distributions, and the role of dyking in extension." (Lines 583-630)

With this information, we provide a brief comparison to strain accommodated on the Exmouth Plateau by other processes:

"Late Jurassic crustal extension by dyking, which we estimate could be up to $\sim$0.33%, was likely much less than that accommodated by Tithonian-to-Valanginian faulting in the lower ($\beta \sim$2.65–2.8) and upper crust ($\beta \sim$1–1.1) across the Exmouth Plateau (cf. Karner and Driscoll, 1999; Rohrman, 2015)." (Line 819)

Given the uncertainties involved in estimating dyke thickness from VZD thickness, and in estimating dyke-driven extension from likely a small proportion of constituent dykes within the swarm (i.e. those imaged by seismic), we have elected not to develop the discussion further or attempt to estimate overpressure. However, we note that work is ongoing examining the dyke-induced faults, which will help constrain extension and therefore allow us to better investigate magma pressure conditions.

20) The error quantification really seems key to what can and can't be said in this study. It appears the VDZ thickness (for example) generally is thicker than dyke thickness, so please state this early on in the paper.

We now mention this in the abstract:

"Borehole data reveal one $\sim$130 m wide VZD corresponds to an $\sim$18 m thick, mafic dyke, highlighting that the true geometry of the inferred dykes may not be fully captured by their seismic expression." (Line 17)

21) What is the composition of these dykes?

Borehole data reveals at least one of the dykes is mafic (see response to comment 20). We have no further constraints on the geometry of this or other dykes.

22) Can you use the general thicknesses of dykes in sedimentary basins to see how the VDZ thickness compares?

See response comment 19.

23) Seems strange to place errors as '+/-' without a bit more explanation.

See response to comment 1 by reviewer 1.

24) It seems likely VDZ thickness overestimates dyke thickness quite substantially? Is it likely to be consistent e.g. as

See response to comment 19.

25) I think something that is missing from the text is a comment about what your study means for reinterpreting datasets where dykes may have intruded and yet can't be imaged? Is there an opportunity to state what proportion of magmatism might be underestimated in relevant comparative regions?

Given limitations in our volume estimates (see response to comment 18), it is difficult to estimate potential 'missing' proportions of magmatism. We also consider that our previous text, dealt with how this study can be used to aid interpretation of dykes in datasets where they may have been missed:

"Our work extends a growing consensus that vertical dykes can be recognised in seismic reflection data imaging continental margins (e.g., Jaunich, 1983; Kirton and Donato, 1985; Wall et al., 2010; Bosworth et al., 2015; Ardakani et al., 2017; Holford et al., 2017; Malehmir et al., 2018; Plazibat et al., 2019). Key criteria for defining vertical dykes in seismic reflection data include: (i) identification of thin, long, tall, typically subvertical zones of disturbance within otherwise sub-parallel reflections defining the host rock (e.g., Figs 6 and 7) (e.g., Wall et al., 2010; Eide et al., 2018; Minakov et al., 2018); (ii) lack of lateral or vertical offset of host rock strata, best revealed by mapping piercing points (e.g., fluvial channels, pre-existing structures) across inferred dyke-like features (e.g., Figs 5 and 8), which suggests the features are not strike-slip or steeply dipping

normal faults; and (iii) potential association with overlying pit craters or dyke-induced normal faults, which are likely easier to resolve and map in seismic reflection data compared to dykes (e.g., Figs 6, 7, 12 and 13). By increasing our collective awareness of how these criteria can be used to identify dykes in seismic reflection data, we expect more dyke swarms will be revealed across continental margins worldwide. Recognition of dyke swarms within seismic reflection data will help us produce better physical models of the subsurface, aiding our understanding of a margins thermal history, and fluid and/or gas plumbing systems of sedimentary basins." (Lines 834-846)

26) Line 8 – 'extend laterally for..'

Corrected.

27) Line 9 – 'their presumed rapid emplacement,..'

Corrected.

28) Line 16 – can you give an indication of the quantification of the detail? To what quantity of resolution?

We have removed the reference here to 'unprecedented detail', partly because it simply served to aggrandize the work and is thus unnecessary, but also because it was erroneous in the sense that field-based studies can examine relatively small parts of a dyke swarm in 3D in more detail.

29) Line 16 – what do you mean the latest? Relative to what?

See response 13 to reviewer 1.

30) Line 17 – the word 'long' needs some context. Is it the horizontal 'length'? The vertical 'length'? Length generally implies to the longest dimension. Are these vertical dykes or blade-like (horizontal) dykes? How deep do they extend? Please give a general overview of the measurements made.

We agree that providing some of the actual data in the abstract would be useful and

have thus added: "Dykes are expressed in our seismic reflection data as ∼335–68 m wide, vertical zones of disruption (VZD), in which stratal reflections are dimmed and/or deflected from sub-horizontal. Borehole data reveal one ∼130 m wide VZD corresponds to an ∼18 m thick dyke, highlighting that the true geometry of the inferred dykes may not be fully captured by their seismic expressions. The Late Jurassic dyke swarm is located on the Gascoyne Margin offshore NW Australia and contains numerous dykes that extend laterally for >170 km, potentially up to >500 km, with spacings typically <10 km. Although limitations in data quality and resolution restrict mapping of the dykes at depth, our data show they likely have heights of at least ∼3.5 km." (Lines 15-21)

31) Line 25 – please define 'rapidly' quantitatively (or at least give a range of timescales).

We have removed the use of 'rapidly' because, in hindsight, it is difficult to properly ascertain the temporal longevity of an entire dyke swarm.

32) Line 27 – 'We recognise' – I think you mean in the literature there are 3 dyke swarm geometries. Please rephrase to take yourselves out of the writing.

Rephrased to:

"There are three principal dyke swarm geometries..." (Line 32)

33) Line 34 – '..emplacement is thought to be primarily..'

Corrected.

34) Line 34 – 'extending the host rock rather than through magmatic overpressure'.

Corrected.

35) Line 36 – 'drive crustal extension, influencing...' – I wonder if this really can be stated based on the observations, or could it be said these dykes are a consequence of crustal extension? Seems the latter if the magma passively fills fractures.

This is a problem we (and others) have faced several times; i.e. how can you define whether dyking drives extension, or extension drives dyking? We agree that if magma is passively filling fractures, then extension is facilitating dyking. However, we would highlight that extension typically occurs via the development of moderately-to-steeply dipping normal faults, rather than formation of sub-vertical tensile fractures. Furthermore, there has been a significant amount of work on the relationship between rifting and dyking in the East Africa Rift showing that dyking actively contributes to extension (see references below). We have modified the text to take this uncertainty into account and to provide further relevant references:

"Their geometry and scale means dyke swarms can thus contribute to crustal extension, influencing plate tectonic processes on Earth and shaping other planetary bodies (e.g., Halls, 1982; Ernst and Buchan, 1997; Ebinger and Casey, 2001; Wilson and Head, 2002; Wright et al., 2006; Paquet et al., 2007; Ernst et al., 2013)" (Line 56)

36) Line 39 – 'syn-emplacement stress conditions' – please clarify that these stresses could be local or regional

We have modified the text to read:

"...also provide a record of local and/or regional syn-emplacement stress conditions..." (Line 49)

37) Line 48 –magnetic surveys provide insight too.

We have modified the text to account for this:

"...Earth's surface or identified in airborne/satellite imagery and remote sensing data" (Line 57)

38) Line 53-54 – Kavanagh and Sparks (2011) data also describes vertical variation in dyke geometry, as well as the lateral variation you attribute.

Our original text was overly simplistic here, so to reflect this important point raised by

the reviewer, we have modified the text to read:

"Integrating these datasets typically emphasises the lateral variability in dyke swarm architecture, although they can show how dyke properties change over vertical distances of hundreds of metres (e.g., Kavanagh and Sparks, 2011). In contrast, seismic reflection data can be used to track changes in dyke swarm structure with depth over hundreds to thousands of metres (Phillips et al., 2018)." (Lines 73-75)

39) Line 54-57, 67-68 - Phillips et al. (2018), Magee et al. (2019) – how is your paper different to these? Please clarify.

We have removed the citation to Magee et al. (2019) in Line 75 (was 54-57). In Line 96 (was 67-68) we mention the importance of 3D seismic reflection data to advancing our understanding of dyke swarms. Whilst Magee et al. (2019) provides a review of Phillips et al. (2018), it also discusses other seismically imaged dykes and highlights the potential future role of seismic reflection data in dyke swarm analysis. We therefore feel it pertinent to keep this citation here.

40) Line 72 – 'length' – this needs some orientation in space. Lateral? Vertical? Not both?

Corrected to 'horizontal length'. See response to comment 17.

41) Line 161 – "controls and is reflected in" – isn't this the same statement repeated? Either 'controls' or 'reflected in' would be sufficient. I don't see the difference.

We have removed 'and is reflected in'.

42) Line 162-163 – "dyke lengths" – I am confused by what you mean by 'length'. Vertical or horizontal dimension? Clarifying this at the start of the paper would be helpful.

Throughout the manuscript we have now clarified measured parameters; e.g., 'length' is always horizontal length.

43) Line 164-165 – "follows a Weibull distribution" – can this point be made as generally as this? These are just two studies you point to, and actually the suggestion is primarily made in the Krumbholtz paper only. Please give some context to the statement. I don't think the statement can be generally made based on just these two papers cited.

We have removed reference to the Weibull distribution here as it is not necessary to raise this in the methodology section.

44) Line 166 – 'dyke geometry and distribution' seems more appropriate than 'dyke properties'

We used 'properties' to avoid using 'distribution' twice in a sentence where each use meant something different (i.e. statistical distribution vs geographical distribution).

45) Line 170 –remove comma 'plan view tip-to-top length (L)'

Corrected.

46) Line 170 –now I understand what you mean by 'length'. I think it would be clearer if you rephrased throughout the paper as 'horizontal length' to be clearer.

See response to comment 17.

47) Line 173 - Ditto with regard to the use of 'width'. In the other datasets you referred to in the literature review this dimension was called 'thickness' so please be consistent with this and rephrase.

See response to comment 17.

48) Line 198 – 'VZD thickness' would be clearer (rather than relating a width to a thickness, relate a thickness to a thickness)

See response to comment 17.

49) Line 323 – "We therefore consider it unlikely that the VZDs are faults." If these dykes have not intruded existing faults, then does this suggest the dykes created there

own fracture during propagation and were overpressured?

See response to comment 35.

50) Line 391 – 'breadth, thickness and spacing'

For clarity and consistency with the rest of the manuscript, we use 'horizontal length' instead of 'breadth'.

51) Line 396 – 'Seismic reflection data thus provide an opportunity to examine and quantify the 3D structure of a dyke swarm independent of the potential bias introduced by the processes (e.g., erosion) controlling how dyke swarms intersect the surface.' I feel this is overstated. It is possible to access different depths of dyke swarms in the geological record by studying areas at different palaeodepths. The thickness increase of dykes with depth based on seismic data seems too speculative at the moment given the errors associated with the method (particularly in that dimension).

We have removed 'unique' to reduce overstatement.

52) 510 – I am not yet convinced that subtle changes in VDZ thickness can be related to dyke thickness changes. The errors appear to be too large.

We agree that data limitations may mean subtle VZD thickness changes are a geophysical artefact, but without further work, we cannot preclude subtle changes are not related to dyke thickness variations. We have rephrased the text to make it clear such an interpretation is speculative: "...(ii) subtle northwards decrease in VZD thickness (Fig. 11B), which we suggest could reflect thinning of dykes, perhaps towards their lateral tip (e.g., Healy et al., 2018)..." (Line 725)

53) 513 – there is ongoing discussion about the Bardabunga dyke swarm and the origin of the magma. The lateral propagation is a hypothesis, however there is also evidence suggesting vertical propagation. So I don't think Bardabunga can be used so decisively to state dykes in your area propagated laterally.

[Figure]

We have acknowledged that the lateral propagation of the Bardabunga dyke is a hypotheses:

"...attained by the BárÃřarbunga-Holuhraun dyke during its possible incremental, lateral propagation..." (Line 726)

We also highlight that we do not rely on a similarity to Bardabunga to support our interpretation that the dykes we examined propagated laterally:

"Lateral propagation of the dykes to the north could be supported by the: (i) the maintenance of dyke upper tip depths (Figs 6, 7, and 11A), consistent with the expectation that horizontally emplaced dykes have fixed upper and lower tip positions (e.g., Townsend et al., 2017); (ii) a subtle northwards decrease in VZD thickness (Fig. 11B), which we suggest could reflect thinning of dykes, perhaps towards their laterally propagating tip (e.g., Healy et al., 2018); and (iii) minor but abrupt changes in the strike of connected dyke segments (Figs 4 and 5), which are reminiscent of the kinked geometry attained by the BárÃřarbunga-Holuhraun dyke during its possible incremental, lateral propagation (Sigmundsson et al., 2015; Woods et al., 2019)." (Line 721)

We have made efforts to emphasise that links to these supporting information are tenuous and open to interpretation.

54) Figure 1 - I like this figure but please remove the 'magma pond' at the base of b-d as it is not needed in your figure and is speculation on the nature of the magma source (suggesting it were entirely molten at once and was a large reservoir, whereas it may also have been transient small batches of distributed melt).

We have modified the figure to remove this.
* * *
Figure 1

(a)

Spanish Peaks

| Dyke swarms | Plume centres | |
|---|---|---|
| ⁄ Linear | ◆ Cenozoic | ● Late Proterozoic |
| ⟋ Radiating | ▼ Mesozoic | ■ Middle Proterozoic |
| ⟩ Circumferential | ★ Paleozoic | ▲ Early Proterozoic |

(b)  (c)  (d)

**Fig. 1.** Fig 1 edit

Figure 5

[Figure]

**Fig. 2.** Fig 5 edit